

# Technical Note: Analysis of concentration-discharge hysteresis loops using Self-Organizing Maps

Arlex Marin-Ramirez, David Tyler Mahoney, Grace McDaniel

Dept. of Civil and Environmental Engineering, University of Louisville, Louisville, KY, 40208, USA

5    *Correspondence to*: Tyler Mahoney (tyler.mahoney@louisville.edu)

**Abstract.** Analyzing concentration–discharge (C–Q) hysteresis loops is essential for understanding both dissolved and particulate constituent sources and transport mechanisms in watershed hydrology. However, traditional hysteresis analysis methods, including loop classification schemes and hysteresis indices, fail to capture the full variability and gradual transitions between loop patterns. To address these limitations, we introduce an alternative approach for characterizing hysteresis patterns in watersheds using the Self-Organizing Map (SOM) algorithm, which better represents loop variability without relying on rigid categories. This technical report outlines the application–and the advantages–of SOM-based hysteresis loop characterization and presents a general workflow for its implementation to characterize C-Q hysteresis for any watershed constituent. We demonstrate the efficacy of the SOM algorithm through a proof-of-concept with sediment transport hysteresis loops. The SOM algorithm was able to classify hysteresis loops with a high degree of accuracy, correctly mapping the amplitude, direction, and concavity of hysteresis loops in the training dataset. We also used the SOM algorithm to develop a *General Turbidity-Discharge (T-Q) SOM*—which may be used as a standardized benchmark for characterizing primary loop types in sediment hysteresis analysis. We demonstrate the use of the *General T-Q SOM* in describing loop frequency distributions and exploring associations with hydrologic variables to infer hydrologic controls of loop types for three watersheds. We found that the *General T-Q SOM* captures key differences in loop shape (and thus sediment transport processes) overlooked by hysteresis indices while preserving the continuum of loop variability lost in classification schemes. Additionally, SOM-based correlation analysis effectively detected associations between loop types and hydrologic variables, enhancing understanding of their hydrologic significance. Combined with high-resolution water quality data, this method offers a powerful tool for advancing the identification of constituent sources and transport mechanisms at the watershed scale. To support broader adoption of the methodology described in this paper, we have developed a Python package, equipped with detailed documentation to facilitate SOM implementation and application in future C-Q analysis.

## 1 Introduction

The analysis of event-scale concentration–discharge (C-Q) hysteresis loops has been employed for decades to infer mechanisms governing the export of dissolved and particulate constituents from watersheds (Mazilamani et al., 2024; Malutta et al., 2020; Liu et al., 2021; Jing et al., 2025; Speir et al., 2024). These hysteretic patterns are commonly used to characterize



time lags between water and constituent exports, flushing versus diluting behavior during storm events, and the dispersion and skewness of water and constituent pulses (Zuecco et al., 2016; Williams, 1989; Evans and Davies, 1998). Hysteresis analyses have proven valuable for identifying sources of distributed pollutants in watersheds (Molder et al., 2015; Pickering and Ford, 2021; Williams, 1989), understanding pollutant transport pathways (Marin-Ramirez et al., 2024; Evans and Davies, 1998; Bettel et al., 2025), improving model calibration (Husic et al., 2023), assessing the impact of watershed and in-stream

alterations such as land use/land cover changes and restoration strategies (Pickering and Ford, 2021; Bettel et al., 2025; Gellis, 2013; Zarnaghsh and Husic, 2021), and supporting better comprehension of catchment functioning to inform management practices (Sherriff et al., 2016; Haddadchi and Hicks, 2021).

Traditionally, the characterization of hysteresis patterns has relied on classification schemes and hysteresis indices (Mazilamani et al., 2024). The former categorizes loops into predefined shapes such as clockwise, counterclockwise, or linear

(Williams, 1989), whereas the latter involves representing loops with an index that typically captures information regarding the loop amplitude and direction (e.g., clockwise or counterclockwise) (Lloyd et al., 2016; Zuecco et al., 2016). Both approaches, however, have limitations. Manual classification is time-consuming and impractical for large datasets, which are increasingly common due to high-resolution in-situ monitoring. While this limitation can be overcome with automatic classification methods (*sensu* Hamshaw et al. (2018)), a more fundamental limitation persists: classification schemes impose

rigid and subjective boundaries that fail to capture the continuous variation in hysteresis loop patterns (see Fig. 1). These strict boundaries result in poor description of loops similarities and differences, potentially hindering statistical analyses that aim to link loop patterns to their controlling factors.

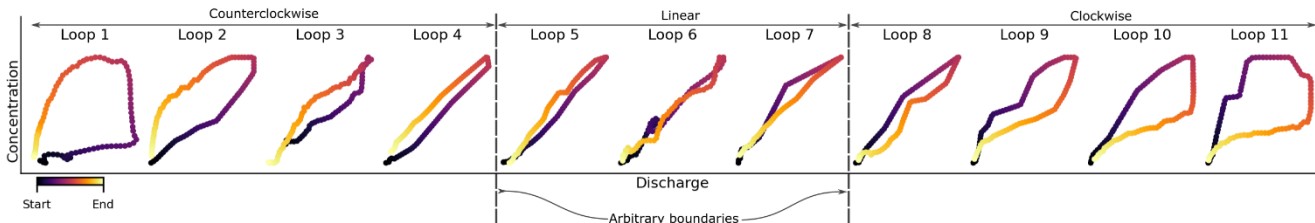

Figure 1. Examples of hysteresis loops arranged to show the smooth transition from wide counterclockwise loops (left),

through linear relationships (center), to wide clockwise loops (right). The figure demonstrates the limitations of traditional classification schemes which rely on rigid boundaries that fail to account for the gradual transition between loop types. For instance, if loops are classified based on the depicted boundaries, the similarity between loop 4 and loop 5 would not be captured. This classification scheme would, in fact, treat loop 4 and loop 5 as being as categorically distinct as loop 1 and loop 5, despite the clear topological similarities. The depicted hysteresis loops are derived from turbidity and discharge data

collected by the USGS across eight different watersheds (see section 3.1.1)

The use of hysteresis indices addresses some of these limitations. They can be automated for large datasets and preserve some of the continuous variability of hysteresis loops. For instance, the indices proposed by Zuecco et al. (2016) and Lloyd et al.





(2016)—which are among the most commonly used—range from -1 to 1, reflecting the transition from counterclockwise to clockwise, with zero indicating the absence of a loop (single line). While the analysis of hysteresis indices has proven effective

for revealing information about dominant hydrologic processes in watersheds (Marin-Ramirez et al., 2024; Zarnaghsh and Husic, 2021; Liu et al., 2021), these indices cannot capture meaningful differences in loop shape. For example, the hysteresis index values (as calculated following Zuecco et al. (2016)) of the clockwise loops in Fig. 2 are equal despite presenting significant qualitative differences, not just in the loop shape itself, but in the characteristics of the discharge and concentration pulses. Particularly, Event 1 and Event 2 differ in their concavity (concave-up and concave-down, respectively) which reflects

differences in the relative spread of the discharge and concentration pulses (Williams, 1989). As can be seen in the lower panel of Fig. 2, concave-up loops occur when the concentration pulse is narrower than the discharge pulse, whereas concave-down loops indicate a wide concentration pulse, likely reflecting a persistent source that remains active during the receding limb of the event. Event 3 on the other hand, shows an extreme case of a leading concentration pulse, where the concentration rises and recedes almost entirely during the rising limb of the hydrograph producing an "L" shaped loop. This loop shape suggests

a strong decoupling between stream discharge and concentration. Despite these differences, the hysteresis index of each loop shown in Fig. 2 is identical, thus demonstrating its limitation to capture loop features relevant for distinguishing transport mechanisms and/or sources.

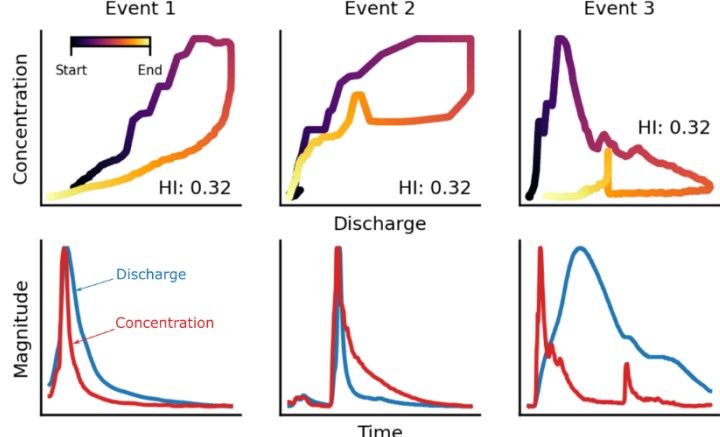

Figure 2. Examples of hydrologic events with hysteresis loops that yield identical hysteresis index values (as defined by Zuecco

et al. (2016)) despite notable differences in their C-Q relationships. For instance, the variation in concavity between events 1 and 2 reflects differences in the width of the concentration pulse, while the distinctive 'L'-shaped loop in event 3 suggests a decoupling between stream discharge and concentration. These examples illustrate the limitations of hysteresis indices in capturing critical differences in loop shapes that are essential for characterizing transport dynamics. The depicted loops are based on turbidity and discharge data collected by the USGS from three watersheds (see section 3.1.1).

To overcome the limitations of traditional classification schemes and hysteresis indices, we propose characterizing discharge-concentration hysteresis loops using Self-Organizing Maps (SOM). SOM is an unsupervised machine learning algorithm





(Kohonen, 1990) widely used for clustering, dimensionality reduction, and data visualization. Although extensively used in water resources analysis (Kalteh et al., 2008; Clark et al., 2020), to our knowledge, its potential for characterizing hysteresis loops has yet to be explored. When applied to C-Q hysteresis loops, SOM generates a two-dimensional map, also called
Kohonen map, that may enable improved representation of loop type variability, deeper exploration of loop characteristics, and enhanced interpretation of diverse C-Q relationships. Furthermore, the algorithm may support key workflows in hysteresis analysis, including data visualization, clustering of similar loop patterns, and correlation analysis to uncover potential controls on loop types, while taking advantage of machine learning's ability to analyze large datasets.

In this technical note, we evaluate the efficacy of the SOM algorithm to represent hysteresis loop patterns and discriminate
hysteresis loop types, compared to traditional classification schemes and hysteresis indices. We use a dataset of turbidity (as a proxy for sediment concentration) and discharge to provide a proof-of-concept for using SOM to discriminate and characterize loop types commonly seen in sediment transport literature. In the following sections, we: (i) present a brief description of the SOM algorithm and explain its advantages for C-Q hysteresis analysis (Section 2), (ii) present a proof-of-concept for the use of SOM to analyze turbidity–discharge hysteresis loops (Section 3 and 4), (iii) compare results from SOM with the hysteresis
index and discuss its efficacy in discriminating samples between different loop types (Section 4), and (iv) show how to use the SOM algorithm with data from any watershed (regardless of the number of hysteresis loop samples available) to describe the distribution of loop patterns and explore the association between loop types and their potential controlling factors (Section 4 and 5). Finally, to support broader adoption of the methodology described in this paper, we have developed a Python package, equipped with detailed documentation to facilitate SOM implementation and application in future C-Q analysis (Section 7).

## 2 Self-Organizing Maps for C-Q hysteresis analysis

### 2.1 Brief description of SOM

The SOM algorithm, developed by Kohonen (1982), is an unsupervised learning technique widely applied in various fields, including clustering, classification, manifold learning, dimensionality reduction, and data visualization (Miljković, 2017). In water resources, SOM has been applied to diverse purposes such as rainfall-runoff modelling, regionalization, clustering of
water quality data, analysis of land use and land cover, and more (Kalteh et al., 2008; Clark et al., 2020). This algorithm generates a discrete representation of a dataset known as a feature map or Kohonen map (Miljković, 2017), which typically takes the form of a two-dimensional grid of nodes arranged in either a rectangular or hexagonal lattice.

Each node in the map has an associated prototype vector that acts as a centroid, representing a cluster of similar samples. Furthermore, the arrangement of prototypes preserves the topological structure of the training data: similar prototypes are
placed close to each other, while dissimilar prototypes are placed farther apart. Conventionally, these prototypes are



represented as n-dimensional vectors, sharing the same dimensionality as the training samples. However, as explained further below, we represent the hysteresis loops using $n \times 2$ arrays which represent a n-length sequence of $(Q, C)$ data pairs.

In the context of hysteresis analysis, the SOM algorithm can extract representative loops from a dataset and arrange them in a coherent manner, offering a two-dimensional, semi-continuous representation of the dataset's loops. By projecting these loops into a two-dimensional space, SOM retains more information compared to the hysteresis index, which offers only a one-dimensional projection, while maintaining a low dimensionality that enables easier analysis and visualization. For instance, the frequency distribution of loop types can be visualized as a heatmap or scatter plot mapped onto the trained Kohonen map. Additionally, this map can serve as a tool to visualize and quantify the relationships between loop types and their potential hydrologic controls, as demonstrated in Section 4.

## 2.2 The training process

Training an SOM begins with defining the number of nodes in the map lattice, which determines the number of prototypes used to describe the input data. While a larger map can represent a dataset with higher accuracy, they are often more difficult to visualize and interpret. Additionally, larger maps may produce prototypes that represent too few or none of the input samples, making the results more susceptible to noise and outliers. Ultimately, defining the map size relies on heuristic rules, domain intuition, and visual examination of the dataset (Vesanto, 2000; Kohonen, 2013). Importantly, however, the optimal size of the map should be refined iteratively, expanding the number of nodes until increases no longer result in meaningful improvements in the quality of the map (Céréghino and Park, 2009). We discuss metrics to assess the map quality in Section 2.3.

Once the map size is defined, initial prototypes must be assigned to each node. Random initialization using samples from the training dataset is commonly used, although more elaborate initialization approaches can also be employed (Attik et al., 2005). With this initial, untrained map, the training process is conducted by feeding random samples sequentially to the map and adjusting the untrained prototypes to match the input data as follows. First, for each sample $X$, the best matching unit (BMU) is identified. The BMU is the node $j$ whose prototype is closest to the input sample as measured by an appropriate *distance function* (e.g., Euclidean, cosine, Dynamic Time Warping, etc.). Next, the prototypes $m^i$ of each node $i$ are adjusted to bring them closer to the input sample using equation (1). The magnitude of the adjustment for each node is controlled by the learning rate (α) and the *neighborhood function* or smoothing kernel ($h^{i,j}$) centered at node $j$ (the BMU for $X$). The most common neighborhood function is the Gaussian Kernel (Kalteh et al., 2008):

$$m_{t+1}^i = m_t^i + \alpha_t h_t^{i,j}(X - m_t^i) \tag{1}$$

This procedure is repeated using all samples in the training dataset until the map stabilizes. During the training process, both the learning rate and radius of influence of the BMU (i.e., the spread of the smoothing kernel) are adjusted using a *decay function*, ensuring their values decrease over time (t). This produces a transition during the training from an initial ordering



and placement phase where prototypes are adjusted to broadly represent the spatial organization of input data, to a fine-tuning phase where prototypes are refined to better represent the input samples (Samarasinghe, 2016).

## 2.3 Quality assessment

Quality measures of Kohonen maps generally assess two map features: *topological preservation* and *quantization accuracy*. Topological preservation ensures that the map maintains the neighborhood relationships of the input space. Hence, a good topological preservation indicates that similar prototypes are placed close to each other and dissimilar prototypes are placed farther apart. Topological preservation can be quantified using the *topographic error*. This error is defined as the fraction of samples whose BMU and second BMU (i.e., the prototype with the second smallest distance to the input vector) are not

neighbors (Kiviluoto, 1996; Pölzlbauer, 2004). Lower topographic errors indicate better topological preservation.

On the other hand, quantization accuracy measures how well the prototypes approximate the input samples and is determined using the *quantization error*. This is defined as the average distance between each sample and its BMU (Pölzlbauer, 2004). A lower quantization error indicates that the prototypes closely resemble the input data. The quantization error can be calculated as a general measure for the map, taking the average over all samples, or as a distributed value over the map, where averages

are calculated over the samples associated with each prototype. The latter offers a more detailed examination of the map quality across prototypes.

The topological error and the quantization error are competing objectives, as decreasing one generally increases the other. Therefore, a balance between both errors must be defined by assessing maps trained with varying hyperparameters such as the number of nodes (map size), the neighborhood spread, and the learning rate. Increasing the number of nodes results in lower

quantization error because a larger number of prototypes can better approximate any given data set. Yet, this comes at the expense of more complex visualization and interpretation. The neighborhood spread has a large influence on both errors: higher spreads increase the radius of influence of each input sample over the map, producing smoother transition between prototypes. This reduces the topographic error but increases the quantization error. Conversely, a narrow neighborhood function isolates each prototype from its neighbors, allowing it to better match its associated samples. This, however, leads to

less gradual transitions between prototypes and larger topographic errors. In broader applications, a trial-and-error approach is often followed where the user must choose the map that better suits their specific dataset (Clark et al., 2020).

A Kohonen map of hysteresis loops can also be evaluated qualitatively since the prototype distribution can be easily visualized. This is especially useful to assess the map's topological preservation which is reflected in a smooth and coherent transition between loop types. Hence, a suitable workflow for map selection could combine an initial assessment based on quantization

and topographic errors, from which an initial subset of candidate maps could be selected. These candidates can then be visually inspected to select the optimum map for further analysis. This workflow is described in more detail in Section 2.5.



**2.4 Quantifying similitude between samples: The *distance function***

The distance function, used to identify the BMU, measures the similarity between two samples. This function directly affects the algorithm's ability to extract the main patterns from the input data and arrange them in a coherent manner. While Euclidean distance is the default selection in most SOM applications, any function that computes the degree of similarity between two samples can be used such as Cosine similarity, Manhattan distance, Minkowski distance, etc. (Samarasinghe, 2016).

Given that a hysteresis loop can be represented as a sequence of (Q, C) data pairs, an appropriate distance function for training an SOM of hysteresis loops is Dynamic Time Warping (DTW). DTW has demonstrated success in clustering and classification across various domains when working with time series data (Ding et al., 2008). Its primary advantage lies in its ability to prioritize the overall shape of the temporal sequences over a strict match of individual data points. This is achieved by dynamically stretching and compressing the time axis to obtain the highest possible alignment between sequences. The Euclidean distance between these aligned sequences is the final DTW distance measure. Hence, DTW distance is upper bounded by the Euclidean distance when no alignment between sequences is possible, but it yields a lower distance when some stretching or compression of the sequences result in a better match, indicating similar shapes. More details on the DTW algorithm and its application in water resources can be found in Lee et al. (2020) or Dupas et al. (2015) and some specifics on its use with two-dimensional time series data can be seen in Shokoohi-Yekta et al. (2017).

Since our objective is to capture similarities between loop shapes regardless of a strict match of individual (Q, C) data points, we consider DTW a suitable candidate for training SOMs for C-Q hysteresis loops. However, additional research is encouraged to better understand how different distance measures influence the trained map and devise tailored distance measures for this specific type of data.

**2.5 Workflow for applying SOM in hysteresis analysis**

A general workflow for utilizing the SOM algorithm for C-Q hysteresis analysis consists of two phases, as illustrated in Fig. 3. The first phase involves training an SOM to represent the spectrum of loop types for a particular dissolved or particulate constituent. To ensure the trained SOM adequately captures all primary loop types for a constituent, we recommend curating a dataset containing samples of all known loop types for the constituent under analysis during the training phase. Additionally—and critical for the objectives of this technical note—curating the training dataset facilitates evaluation of the algorithm's ability to represent and distinguish hysteresis loops. In practice, curating the training dataset also mitigates bias toward more frequently occurring loop types in natural systems by incorporating a balanced number of samples for each type. We discuss some advantages and limitations of this approach more in Section 5. During the training process, an SOM should be selected that provides a coherent arrangement and smooth transition between loop types (i.e., low topographic error) while accurately representing the training data (i.e., low quantization error).



The second phase of the workflow consists of using the trained SOM to characterize loop types from any dataset (assuming that they are of the same constituent of interest) to aid with analyzing hysteresis patterns in watersheds. This involves mapping each hysteresis loop in the dataset to its corresponding BMU within the trained SOM. Importantly, this procedure does not

require reapplying the SOM algorithm itself; rather, it only requires the distance function to map each sample to its corresponding prototype. As demonstrated in our proof-of-concept in section 3, this approach can be employed to characterize the frequency distributions of loop types and explore associations between loop types and hydrological variables, thereby providing valuable insights into the underlying controlling mechanisms.

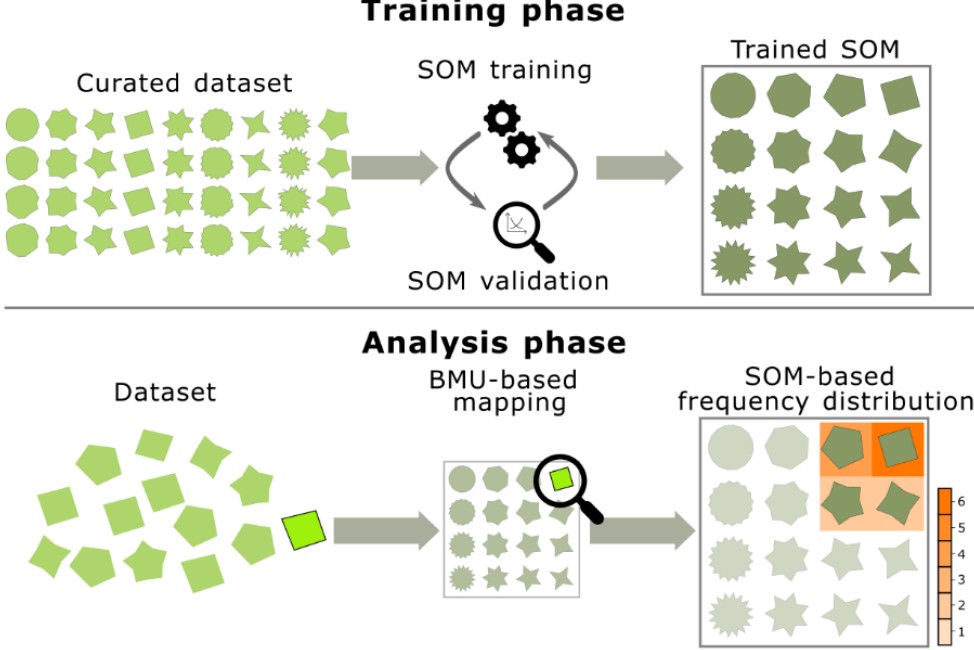

Figure 3. Workflow to generate an SOM for C-Q hysteresis loops (training phase) and apply the SOM for C-Q hysteresis analyses in watersheds (analysis phase). Here, we illustrate the generation of the SOM using different shapes, which are analogous to the hysteresis loop types that might be found for a dissolved or particulate constituent in a watershed. In the bottom panel (analysis phase), we demonstrate how hysteresis loops from a new dataset get mapped to the trained SOM, where the shade of orange represents the frequency with which the shape occurs in the dataset.

## 3 Applying SOM to turbidity-discharge hysteresis loops


We employ turbidity-discharge (T-Q) data as proof-of-concept to demonstrate the proposed workflow for training and applying SOM in C-Q hysteresis analysis. Although sediment is the central focus of the current study, we stress that this workflow is adaptable to other constituents such as nutrients, dissolved solids, metals, and more. We encourage future studies to expand on this approach to explore its application across a broader range of constituents. The following sections detail the

implementation of this workflow.



### 3.1 Training phase

### 3.1.1 Data curation and preprocessing

We surveyed the sediment transport literature to compile the primary loop types included in the curated dataset (Fig. 4). Our review indicates that the majority of sediment hysteresis loop types recognized in the literature were originally identified by

Williams (1989), with the *Figure L* loop introduced by Hamshaw et al. (2018). Additional studies have introduced nuanced variations of these loop types; for example, Bettel et al. (2025) introduced the "J" loop for a karstic system in Kentucky, USA which resembles concave-up loops. However, our analysis suggests that sediment hysteresis loop diversity can largely be explained by the 17 loop types identified in Fig. 4, encompassing *single-line*, *Figure 8*, *clockwise*, and *counterclockwise* topologies. Complex loops were excluded from the curated dataset due to their irregularity and lack of a standardized

classification system. While additional loop types may exist beyond those included, they are generally infrequent and relevant only to specific studies or watersheds. We discuss the incorporation of less common loops in Section 5 and encourage the development of tailored SOMs to accommodate specialized applications.

The curated dataset consists of 20 samples for each loop type, resulting in a total of 340 loops. Loops were manually delineated using publicly available discharge and turbidity data collected at 15-minute intervals by the USGS across 37 stream gauges in

the United States. Data were retrieved from the National Water Information System using the Data Retrieval Python package (Hodson et al., 2023). The Supplementary Information provides detailed site information and data periods (Table S1), while the complete set of loops is presented in Figs. S1 and S2.

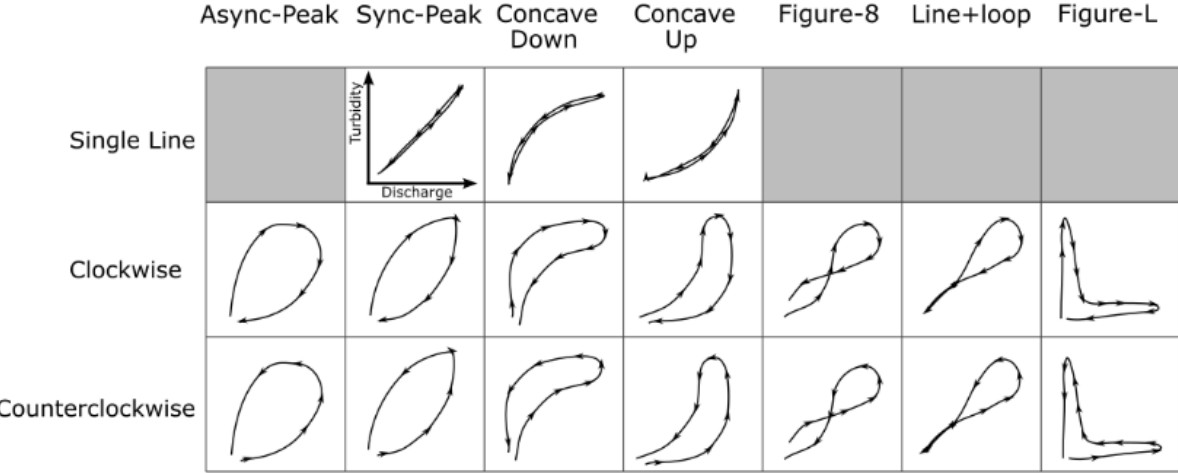

Figure 4. Loop types considered in the training dataset. All loop types, with the exception of *Figure-L* loops, were initially
described by Williams (1989). *Figure-L* loops were introduced by Hamshaw et al. (2018)



The SOM algorithm requires samples to be represented as arrays with an equal number of elements. Given that loops in the dataset consist of variable-length sequences of (Q, T) data pairs, we implemented a preprocessing procedure to convert all loops into sequences of equal length, ensuring compatibility with the SOM algorithm. This procedure is illustrated in Fig. 5 and involves the following steps. First, we applied a moving median with a 5-point window to the turbidity values to reduce the influence of outliers (Fig. 5b). Second, the Q-T data were scaled to a [0,1] interval using min-max normalization based on the minimum and maximum values for each event. This scaling, commonly applied prior to the calculation of hysteresis indices, facilitates comparison between loops of different magnitudes (Lloyd et al., 2016); and third, we interpolated the resulting variable-length, scaled T-Q data (Fig. 5c) to produce equal-length sequences (Fig. 5d) of 100 data pairs. The interpolation ensures that data points are equally spaced in the Q-T plane.

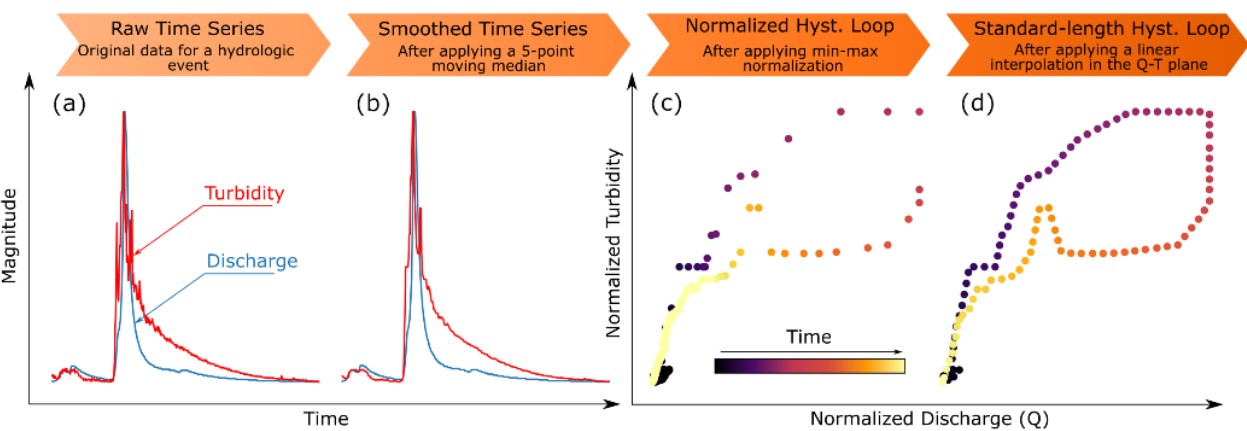

Figure 5. Workflow illustrating data preprocessing steps: a) original discharge and turbidity data. b) A 5-point moving median was applied to turbidity data to mitigate outliers. c) Q-T data were min-max normalized to a [0,1] interval. d) the scaled, variable-length data were interpolated into equal-length sequences of 100 data pairs.

### 3.1.2 SOM training and validation

The map training process involves three key steps: map size selection, map topology selection, and map refinement (Samarasinghe, 2016).

During the map size selection step, SOMs ranging from 5x5 to 13x13 nodes were trained using a grid search approach, varying hyperparameters such as neighborhood spread (0.5–13) and learning rate (0.05–0.9) across five epochs, resulting in approximately 900 trained maps. Highly distorted maps, identified by topographic error, were excluded. The elbow method was then applied to determine the optimal map size, where increasing the number of nodes no longer significantly reduced quantization error (Nainggolan et al., 2019).

Once the optimal map size was identified, we evaluated a subset of maps to find the one that best captured and organized the spectrum of loop types in the dataset. This subset was chosen through Pareto-optimal analysis, identifying maps on the frontier





of topographic and quantization errors. The general arrangement of loop types was assessed based on our conceptual

understanding of similarities and differences between loop types to ensure smooth transitions between similar loop types and clear separations between contrasting ones. For example, maps were ranked higher when clockwise and counterclockwise *Figure L* loops were placed farther apart than clockwise and counterclockwise *Sync-Peak* loops (see Fig. 4), as the former loop shapes indicate more pronounced time mismatches between discharge and turbidity.

In the final refinement step, the selected map underwent retraining for an additional five epochs. While the map from the

preceding step focused on optimizing the overall arrangement of loop types, this step targeted enhanced quantization accuracy. To achieve this, the map was retrained using a reduced neighborhood spread, updating only the Best Matching Unit (BMU) during each iteration. This adjustment significantly improved the alignment between each BMU and its associated samples. A constant learning rate of 0.05, with no decay, was maintained throughout this refinement.

### 3.1.3 Map quality evaluation

The final evaluation step aimed to assess the SOM's ability to represent the full spectrum of loop types included in the training dataset. Each loop was mapped to its corresponding prototype (i.e., BMU) and underwent a visual examination to assess how loops within a given class aligned with their prototypes. Specifically, we evaluated the consistency of loop types mapped to a given prototype, as well as the resemblance of individual loops to their BMU. Additionally, we evaluated the extent to which loops from the same or closely related classes (e.g., *async-peak* and *sync-peak;* see Fig. 4) were mapped to neighboring

prototypes indicating the SOM's ability to recognize shared shape characteristics in alignment with conceptual classifications. This evaluation approach highlights the features that the SOM algorithm effectively captures while identifying those requiring more nuanced analysis. Since the SOM training process is fully unsupervised, it disregards manually assigned labels, relying solely on loop shape to detect similarities and differences. We note that while the curated dataset was manually labeled (to aid with training the model on a balanced dataset), these labels are not seen by the model. Finally, we chose to avoid quantitative

classification metrics, such as confusion matrices, to evaluate the ability of the SOM to characterize hysteresis loops, as they would require converting the SOM mapping into a nominal classification scheme—undermining SOM's key advantage of preserving gradual transitions between loop types.

### 3.2 Analysis phase

To demonstrate the application of the trained SOM for analyzing hysteresis patterns, we collected a secondary dataset

consisting of T-Q hysteresis loops from three monitoring stations. The coordinates of these stations and key properties of their associated watersheds are provided in Table 1. This dataset supports two primary analyses for characterizing hysteresis patterns: (1) exploring the frequency distribution of loop types within a watershed, and (2) identifying associations between loop types and hydrologic variables.





Table 1. USGS monitoring stations and watershed properties used in the SOM-based analysis phase.

| USGS code | | 07364130 | 03254480 | 03289000 |
|---|---|---|---|---|
| **Latitude** | | 33.96 | 38.84 | 38.04 |
| **Longitude** | | -91.69 | -84.53 | -84.63 |
| **Watershed Area (km²)** | | 311 | 47 | 62 |
| **Watershed Mean Slope (m/m)** | | 0.012 | 0.13 | 0.08 |
| **Watershed Urban Area (%)** | | 2.03 | 1.31 | 15.8 |
| **Mean Discharge (m³ s⁻¹)** | | 7 | 0.85 | 1.2 |
| **Soil Texture** | **Sand (%)** | 9 | 6 | 6 |
| | **Silt (%)** | 50 | 61 | 64 |
| | **Clay (%)** | 41 | 33 | 30 |


Hydrologic events for watersheds 07364130, 03254480, and 03289000 were manually extracted from 15-minute discharge and turbidity data provided by the USGS and retrieved via the National Water Information System. For station 03289000, turbidity data were collected by the University of Kentucky using a YSI 6-series optical turbidity sensor. All loops underwent preprocessing consistent with the curated dataset (see Section 3.1.1). This process resulted in 27, 54, and 70 events for gages

07364130, 03254480, and 03289000, respectively. Additional hydrologic variables were included for watershed 03289000, including rainfall 15 hours prior to events, average discharge over the 5 preceding days, and the ratio of old-water to event-water, derived from our previous work in this watershed (Marin-Ramirez et al., 2024). These variables serve as proxies for rainfall, antecedent moisture conditions, and dominant hydrologic pathways associated with each event, respectively.

For each site, loop frequency distributions were generated by mapping loops to their corresponding SOM prototypes and

visualizing them as heatmaps. Distributions for 07364130 and 03254480 were compared to those obtained using the hysteresis index proposed by Zuecco et al. (2016). For watershed 03289000, the trained SOM was employed to explore relationships between loop types and hydrologic variables. This station was selected based on prior understanding of the hydrologic processes controlling sediment transport in the watershed (Marin-Ramirez et al., 2024), providing a framework for validation and comparison with the trained SOM. Associations between hydrologic variables and loop types were initially explored

through visual analysis, where median values of the variables were plotted across SOM prototypes as heatmaps to reveal patterns linking high or low values with specific loop types.

To quantitatively assess these associations, a correlation approach was applied. First, BMU coordinates were transformed into a two-dimensional representing each loop type. This two-dimensional index served as the predictor variable in a multiple linear regression model. Rank-normalized hydrologic variables were used as predictands, allowing the model to capture non-linear,

monotonic relationships. However, non-normalized variables could also be employed. Correlation coefficients were used to evaluate the strength of associations, while the regression plane's coefficients represented gradients showing the direction of



maximum change in the hydrologic variable. These gradients helped identify loop types associated with higher or lower values of the variables. Results were visualized using a biplot chart, enabling simultaneous exploration of relationships between loop types and multiple hydrologic variables.

## 4 Results

### 4.1 SOM training

#### 4.1.1 Balancing quantization and topographic errors

The elbow method identified 8×8 nodes as the optimal map size, where quantization accuracy stabilizes (Fig. 6a). While larger maps may achieve lower quantization errors, this improvement comes at the cost of increased distortion, reflected by high

topographic errors. Moreover, visual inspection of maps with varying sizes (not shown) revealed no advantages in terms of the number of loop patterns represented.

The pareto frontier of these 8×8 maps show the expected inverse relationship between topographic and quantization errors (Fig. 6b). After visually examining six maps along this frontier, the map with the best overall distribution of loop types was selected (black dot shown in Fig. 6b). We favored maps with clear separations and smooth transitions between clockwise and

counterclockwise loops and between concave-up and concave-down loops, which are representative of both the timing and duration of sediment transport during events. The selected map exhibited a low topographic error (0.006) but a relatively high quantization error (0.73). However, the quantization error was improved during the refinement phase. The final map achieves a well-balanced trade-off between topological preservation (TE = 0.04) and quantization accuracy (QE = 0.66).

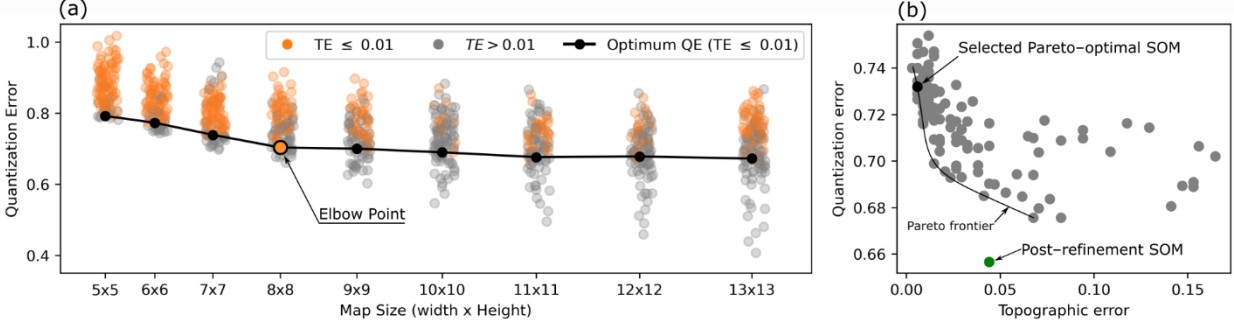

Figure 6. a) Quantization Error (QE) across varying map sizes, illustrating the expected decrease in QE as map size increases. While larger maps continue to lower the minimum QE, this reduction stabilizes when considering maps with low Topographic Error (TE ≤ 0.01). Similar stabilization patterns are observed across other TE thresholds (0.05 and 0.1). b) Scatterplot displaying QE and TE for 8×8 maps. The map selected after visual inspection is highlighted, alongside the final QE and TE following map refinement training.





### 4.1.2 A trained SOM for sediment hysteresis analysis

Fig. 7 shows the trained SOM generated using the curated dataset, which may be used for sediment hysteresis analysis. The map provides an organized representation of most loop types from the curated dataset, with smooth transitions between different classes. In general, the map arranges loop shapes based on the loop amplitude, rotational direction (clockwise or counterclockwise), and concavity following the directions defined by the diagonals. These diagonals can be conceptualized as axes of a Cartesian plane: the diagonal running from the lower left to the upper right (LL-UR) represents variations in loop amplitude and rotational direction, while the diagonal from the upper left to the lower right (UL-LR) represents changes in concavity.

The LL-UR diagonal illustrates the expected progression between clockwise and counterclockwise loops as illustrated in Fig. 1. Wide clockwise and counterclockwise loops occupy the extreme ends of this diagonal, while single lines are positioned near the center of the transition. *Figure L* loops notably appear at opposite ends of this transition, emphasizing their role as extreme cases of clockwise and counterclockwise loops, as they represent a pronounced mismatch in concentration between the rising and falling limbs (Fig. 2). The UL-LR diagonal, on the other hand, captures the transition from concave-up to concave-down loops. Hence, all concave-up (*clockwise*, *counterclockwise* and *single lines*) loops are placed in the upper-left quadrant of the map, while concave-down (*clockwise*, *counterclockwise* and *single lines*) occupy the lower-right quadrant.

Furthermore, these diagonals effectively serve as symmetrical axes, highlighting the map's coherent and robust topological structure. For example, clockwise loops along the LL-UR diagonal (e.g., H1, G2, F3, E4) are mirrored by their counterclockwise equivalents (e.g., D5, C6, B7, A8).

Interestingly, the map also reveals the presence of loop types not explicitly considered during data curation. For instance, it distinguishes between concave-up and concave-down clockwise *line+loop* loops (e.g., prototypes C1 and H6), underscoring its ability to capture differences in concavity. On the other hand, *Figure 8* loops are less accurately represented—they appear as narrow loops placed close to single lines (e.g., D4, E5). This suggests that the map may have less discriminatory power for *Figure 8* loops.



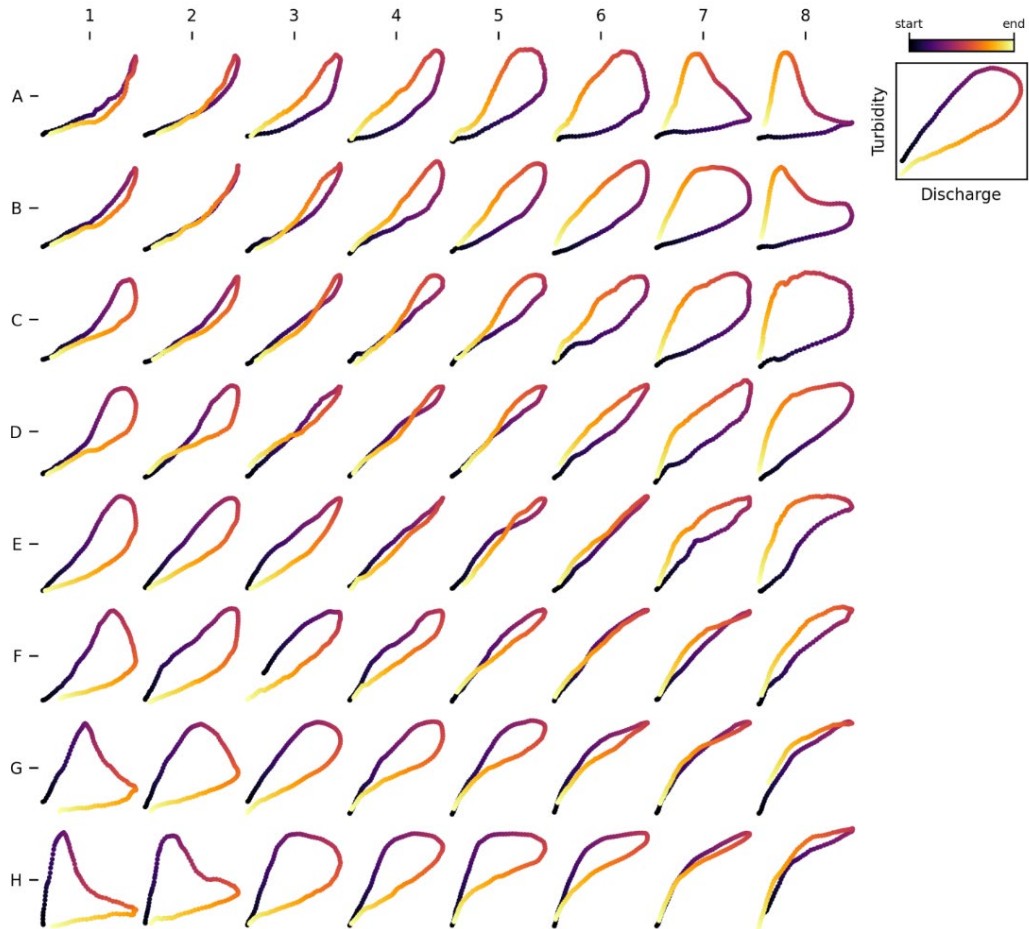

Figure 7. Kohonen map for Turbidity – Discharge hysteresis analysis. The map was trained using a curated dataset comprising 17 different loop types, with 20 samples for each loop type. We refer to this as the *General T-Q Map*.

### 4.1.3 Map evaluation

To evaluate the efficacy of the Kohonen map in discriminating between different loop types, we mapped each loop in our curated dataset to its corresponding prototype (i.e., its BMU). As shown in Fig. 8, the results are consistent with our manual classifications, noting that the manual classification was not seen by the model as part of the training process. In particular, the map effectively distinguishes rotational direction (clockwise, counterclockwise), loop amplitude (ranging from wide loops to single lines), and concavity. However, the self-intersecting structure of *Figure 8* loops is less effectively captured.

*Single-line* loops are consistently mapped to prototypes that best represent their shape along the UL-LR diagonal (Fig 8a), while *Figure L* loops are properly mapped to their corresponding prototypes at the map's corners (Fig 8b, 8c). The spectrum of clockwise and counterclockwise loops (including *Async-Peak*, *Sync-Peak*, *concave-down*, *concave-up* and *line+loop* loops)



is also well-represented, with clockwise loops occupying the lower-left quadrant and counterclockwise loops in the upper-left quadrant of the map (Fig 8b, 8c).

Overlap exists between some loop types, which is expected due to the similar topologies of many loops (see Fig. 1). For instance, several counterclockwise *Async-Peak* and *Sync-Peak* loops map to prototype B6 (Fig. 8c), while their clockwise counterparts group under prototype E2 (Fig. 8b). A closer examination of these clusters (Fig. 9) reveals that they comprise of loops sharing similar characteristics in terms of direction, amplitude, and concavity—or the lack thereof—demonstrating the algorithm's effectiveness in discriminating loops based on these features.

*Figure 8* loops, in contrast, exhibit a more dispersed mapping, often associated with prototypes that lack a clear *Figure 8* resemblance. Nonetheless, the mapping is not entirely flawed. Closer examination reveals that *Figure 8* loops are matched to prototypes resembling their loop amplitude, direction, and concavity (see Fig. 9). Note that in our manual classification, these *Figure 8* loops were categorized as clockwise based on the rotational direction of the second loop (i.e., the one with larger C-Q values). Nevertheless, they were mapped to counterclockwise loops following the rotational direction of the dominant loop, which in this case is the first loop. Despite this apparent contradiction, we consider this an acceptable representation of these loops.

Overall, we consider the trained map to perform satisfactorily when representing and discriminating between a broad spectrum of loop patterns associated with sediment transport, as informed by our survey of the sediment transport literature. As such, we refer to the trained map as the *General T-Q SOM*. We discuss some specific uses of the *General T-Q SOM* for broader hysteresis analyses in Section 5.









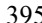

Figure 8. The figure shows the mapping of all samples from each loop type (see Fig. 3) in the curated dataset to their respective Best Matching Unit (BMU) on the Kohonen map. To enhance clarity, loop types and their corresponding mappings are divided
into four distinct groups. SP: *Sync-Peak*, AP: *Async-Peak*, CU: *Concave-Up*, CD: *Concave-Down*, ccw: *Counterclockwise*, cw: *Clockwise*, fL: *Figure-L*, f8: *Figure 8*





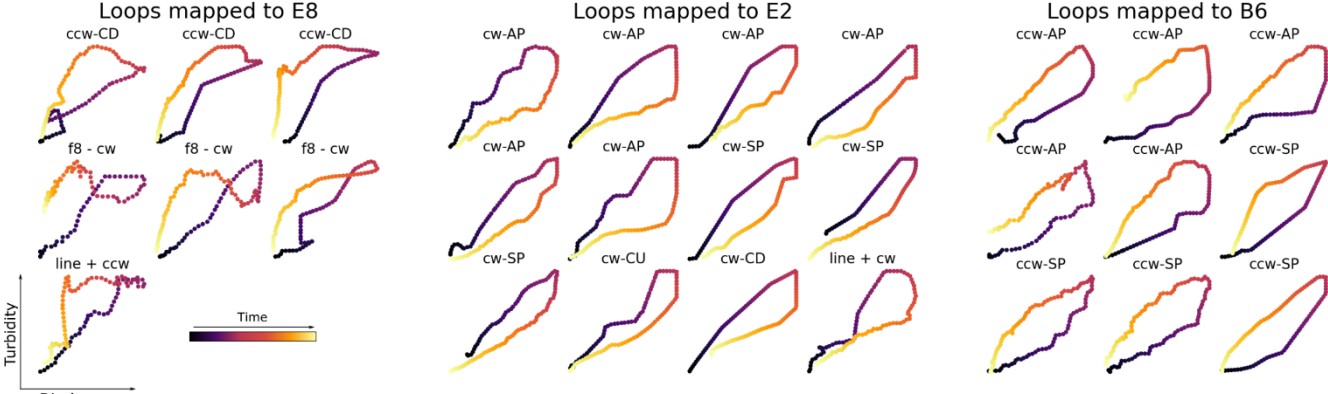

Figure 9. Loops mapped to prototypes E8, E2 and B6 along with their respective classes as defined in the manual classification. While loops from different classes in the curated dataset are mapped to the same prototype, this figure underscores their similarities in terms of loop direction, amplitude, and concavity—or the lack thereof. This highlights the inherently fuzzy boundaries that exist between certain loop types. SP: *Sync-Peak*, AP: *Async-Peak*, CU: *Concave-Up*, CD: *Concave-Down*, ccw: *Counterclockwise*, cw: *Clockwise*, fL: *Figure-L*, f8: *Figure 8*

## 4.2 Analysis of hysteresis loops using the *General T-Q SOM*

### 4.2.1 Characterizing frequency distributions of hysteresis loop types

Fig. 10a shows the frequency distributions of hysteresis loop types found in watersheds 03254480 and 07364130 as heatmaps over the *General T-Q SOM,* contrasted with Fig. 10c, which shows the distributions of hysteresis index. While both watersheds predominantly exhibit clockwise loops, the *General T-Q SOM* reveals additional insights that the hysteresis index distribution cannot capture. Namely, watershed 03254480 shows more concave-down loops, while watershed 07364130 primarily shows concave-up loops. Importantly, these differences are overlooked by the hysteresis index, as its frequency distributions are statistically indistinguishable (two-sided Smirnov-Kolmogorov test, *p= 0.997*).

Fig. 11 reinforces this by illustrating the distribution of hysteresis index values across the map's prototypes, confirming that the hysteresis index does not account for loop concavity. For instance, prototypes along the diagonal from E1 to H4 exhibit identical hysteresis index values, despite a noticeable transition from concave-up to concave-down loops. This highlights the *General T-Q SOM's* ability to capture subtle variations in loop characteristics that the hysteresis index fails to detect.



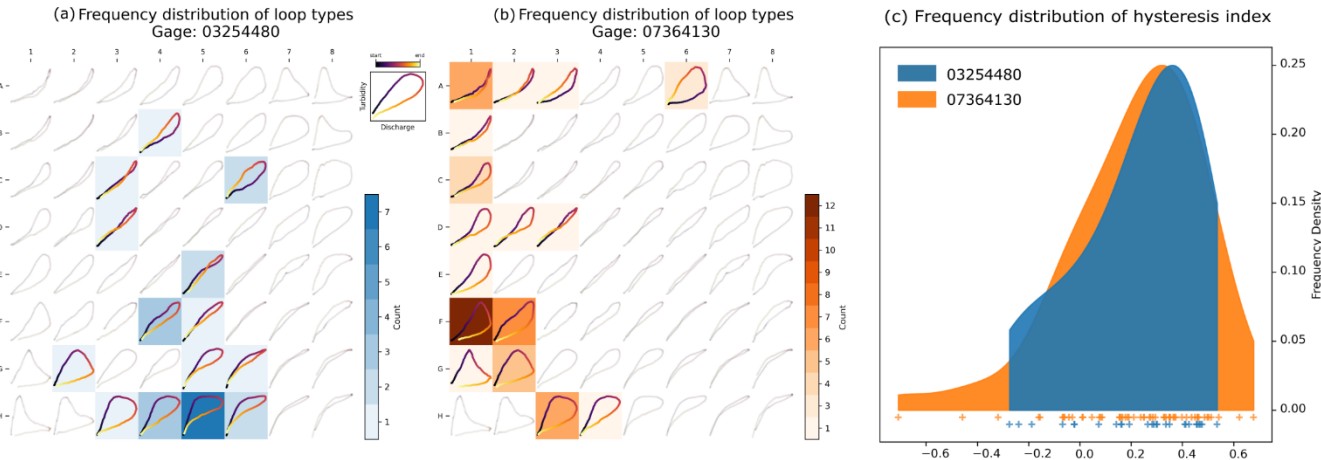

Figure 10. Frequency distribution of loop types for watersheds 03254480 (10a) and 07364130 (10b) mapped onto the *General T-Q SOM* and the hysteresis index (10c). The frequency distributions mapped onto the *General T-Q SOM* reveal distinctions in loop types that are not captured by the hysteresis index.

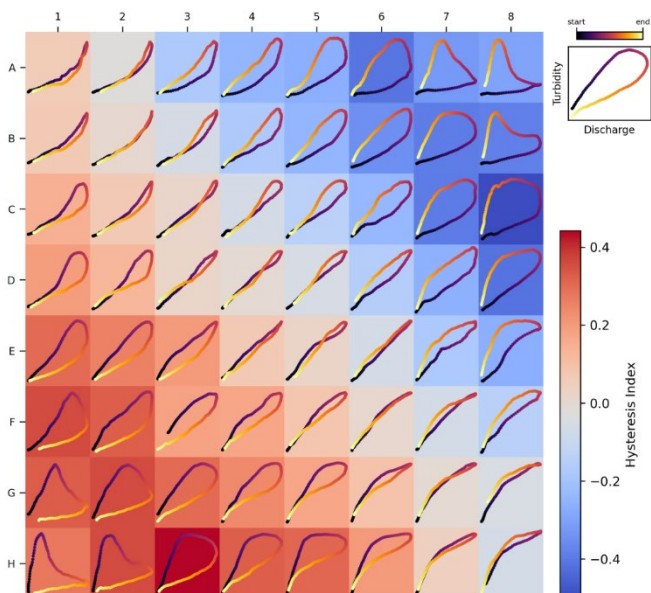

Figure 11. Distribution of hysteresis index values across the Kohonen map's prototypes

Although a detailed analysis of the factors driving differences in loop types between watersheds 03254480 and 07364130 is beyond the scope of this report, the variations in frequency distribution displayed in Fig. 10a and 10b likely reflect differences in sediment transport mechanisms. Concave-down loops are typically associated with significantly higher sediment





concentrations during the falling limb of the hydrograph compared to concave-up loops (see Fig. 2; Williams (1989)). This could be the result of differences in the watersheds's physiographic attributes. Gage 03254480, located in northern Kentucky at the outlet of a 47 km² watershed, is situated in hilly terrain with few flat areas (Mcgrain and Currens, 1978). In contrast, Gage 07364130, located in Arkansas at the outlet of a 311 km² watershed, is characterized by low-relief topography dissected by meandering rivers and streams, typical of the alluvial plain of the Arkansas and Mississippi Rivers. The hydrologic regime

of 07364130 is likely more base-flow dominated than that of 03254480, a condition that has been linked to the formation of concave-up loops in previous research (Bettel et al., 2025). While this explanation remains speculative and requires further analysis for validation, the *General T-Q SOM* proves valuable in identifying variations in hysteresis loops that might have been overlooked with traditional hysteresis indices.

**4.2.2 Identifying associations between hydrologic variables and loop types**

The relationship between loop types and associated variables is shown in Fig. 12, as heatmaps of median values for rainfall (Fig. 12b), antecedent discharge (a proxy for antecedent moisture conditions; Fig. 12c), and the old-water to event-water ratio (a proxy for event flow pathways; Fig. 12d) for watershed 03289000. The heatmaps show clear relationships between loop types and hydrologic variables: precipitation increases from the upper-right to the lower-left corner of the map (Fig. 12b),

linking higher precipitation to clockwise loops and lower precipitation to counterclockwise loops. Similarly, old-water to event-water ratios decrease along this same gradient, indicating a greater dominance of old-water contributions in counterclockwise loops compared to clockwise loops (Fig. 12d). In contrast, antecedent discharge values appear randomly distributed, showing no discernible pattern (Fig. 12c). These findings are consistent with our previous study (Marin-Ramirez et al., 2024), which identified rainfall and old-water to event-water ratios as key predictors of hysteresis variation, while

antecedent conditions showed no significant association with the hysteresis patterns.

Fig. 13 complements this analysis by showing the quantitative relationship between loop types (as represented by BMU coordinates) and the hydrologic variables. The correlation analysis shown in Fig. 13, as indicated by arrow lengths representing coefficients of determination near 0.5, highlights the association between loop type and both precipitation and old-water to event-water ratios. In contrast, antecedent discharge shows negligible association, reflected in a near-zero coefficient and short

arrow. The arrows reveal the direction of these associations: precipitation increases along the transition from counterclockwise to clockwise loops, while old-water to event-water ratios increase in the opposite direction.

While verifying the causation of these relationships is outside the scope of this technical note (and is explored more in Marin-Ramirez et al. (2024)), we emphasize that the method provides a quantitative and repeatable approach for visualizing relationships between watershed variables and loop types, which overcomes limitations of traditional hysteresis anlayses.






Figure 12. Heatmaps for watershed 03289000 of a) loop types, b) precipitation, c) antecedent discharge, and d) old-water to event-water ratio. The figure illustrates a visual approach for analyzing associations between hydrologic variables and loop types. In Figs. 12b -12c, the color intensity represents the median values of the variable calculated for loop samples mapped to each prototype. Consistent patterns are evident for precipitation (higher precipitation for clockwise loops) and the old-water



to event-water ratio (higher ratios for counterclockwise loops), whereas no discernible pattern can be seen for the 5-day antecedent discharge.

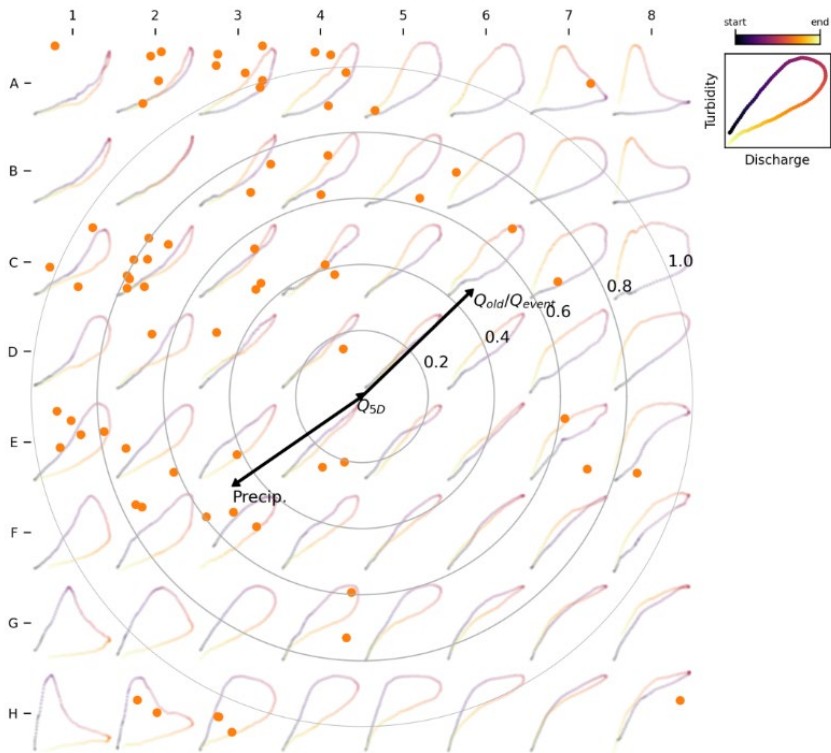

Figure 13. SOM-based biplot. The figure illustrates results from a correlation analysis between hysteresis loop types and hydrologic variables. The direction and length of the arrows are derived from multiple linear correlations, where BMU

coordinates serve as the independent variables and the hydrologic variable of interest as the dependent variable. Arrows indicate the direction of increasing variable values (e.g., precipitation increases for clockwise loops and decreases for counterclockwise loops). The length of each arrow corresponds to the coefficient of determination for the correlation, with reference values provided by the concentric circles. Each dot represents a sample loop, positioned randomly within the square circumscribed by its respective BMU.

## 5 Discussion

### 5.1 Efficacy of the SOM algorithm for C-Q analyses

In this technical note, we introduce a novel approach for characterizing C-Q hysteresis loops using the unsupervised machine learning algorithm, Self-Organizing Maps (SOM). While Hamshaw et al. (2018) previously proposed a machine learning-based method for classifying hysteresis loops, their approach closely mirrors manual classification by assigning loops to nominal categories. As discussed in this study, such methods overlook the continuum of hysteresis loop variability by imposing



arbitrary boundaries on loop classifications, disregarding the nuanced similarities and differences between loop types (see Fig. 1). In contrast, the SOM-based approach presented here preserves the gradual transitions between hysteresis classes, providing a more robust framework for exploring the hydrologic significance of diverse loop types (Fig. 7).

Our method also addresses key limitations inherent to hysteresis index-based approaches, which simplify loop shape variability into a single dimension (e.g., clockwise vs. counterclockwise). By contrast, SOMs offer a two-dimensional framework that enables a more nuanced discrimination of loop types while maintaining low dimensionality to facilitate further analyses, including visualization (see Figs. 10, 12, and 13). Moreover, the flexibility of this approach allows for extensions into higher dimensions, making it adaptable to the specific requirements of diverse applications.

By applying SOM to a curated dataset of sediment transport hysteresis loops representative of sediment hysteresis types
surveyed from the literature, we developed the *General T-Q SOM*. Beyond the advantages over nominal classification schemes, this map improves upon hysteresis indices by capturing changes in concavity—a feature entirely overlooked by traditional hysteresis indices (e.g., Zuecco et al. (2016)). While loop concavity has historically received less attention in hysteresis analysis, previous studies highlight its importance in characterizing C-Q relationships. While Williams (1989) first described concave up and concave down loops, Evans and Davies (1998) demonstrated how relative contributions from different flow
pathways shape distinct concavity patterns, suggesting concavity as an indicator of constituent sources. Bettel et al. (2025) further reinforced this concept, associating concave up loops with higher base flow contributions. These findings underscore the importance of concavity in understanding C-Q dynamics (see also Fig. 2) and highlight the *General T-Q SOM*'s value in supporting hysteresis workflows, particularly for identifying critical sources and pathways of sediment transport.

5.2 Future use of the SOM algorithm for C-Q analyses

By offering an improved characterization of the spectrum of sediment hysteresis loop types observed in watersheds, the SOM algorithm has the potential to deepen our understanding of the factors driving the emergence of varying loop types, including hydrological variables and watershed structural properties, as demonstrated here with the *General T-Q SOM*. Scaling this analysis across watersheds with diverse physiographic conditions is expected to provide valuable insights into the links
between loop types, sediment sources, and transport pathways, offering critical information to mitigate the negative impacts of excessive sediment transport.

While the *General T-Q SOM* (Fig. 7) derived herein was created as a proof-of-concept of applying the SOM algorithm for hysteresis analyses, the map can readily be used for hysteresis workflows more broadly. We demonstrate how this can be done by using data from three watersheds that were not utilized during the map training process. Importantly, for these watersheds,
it was unnecessary to retrain the *General T-Q SOM* to facilitate the analysis.

However, studies examining less-common loop patterns or focused on specific loop features may require training alternative T-Q SOMs. For example, *Figure 8* loops cannot be reliably distinguished from non-figure 8 loops with similar amplitude and concavity by the *General T-Q SOM* derived herein (see Fig. 8). Attempts to improve the representation of *Figure 8* loops (not shown) revealed that including this typology often resulted in highly distorted maps, which compromised the topological



structure and disrupted smooth transitions between loop types. Furthermore, how to incorporate complex loops into the SOM framework remains a challenge. Complex loops do not have repeatable and consistent patterns that can readily be grouped to a single hysteresis prototype.

Addressing these challenges – initially – requires identification of loop types that do not fit well with the *General T-Q SOM*. To determine these events, we propose that two approaches may be employed. For small datasets, visual inspection is

recommended to flag loops that deviate fundamentally from those in the *General T-Q SOM*. This process can leverage the prototype arrangement within the *General T-Q SOM* to generate a structured visualization of the dataset, as illustrated in the supplementary material (Fig. S2). Flagged loops can then be analyzed as an additional category, or the map could be retrained incorporating the additional loop type in the training dataset. For larger datasets, where visualization of the entire dataset is impractical, outlier detection methods can be easily implemented for SOMs, for example, by comparing the QE of individual

samples with the distribution of QEs for their associated prototypes as proposed by Munoz and Muruzábal (1998). Interestingly, identified outliers may serve as candidates for training additional SOMs, aiding in the discovery of novel hysteresis patterns. We encourage researchers to use the *General T-Q Map* as an initial benchmark and point of comparison to aid with uncovering such patterns in future analyses, thereby facilitating new understanding of the controls of sediment transport in watersheds.

Many additional analyses can be facilitated by applying the SOM algorithm to hysteresis patterns. For example, the SOM algorithm can facilitate investigation into seasonal variations in sediment hysteresis patterns in individual watersheds, such as differences between wet and dry seasons. Furthermore, the SOM algorithm can support regional or continental-scale characterizations of hysteresis behaviors, enabling comparison of hysteresis patterns across climate zones (e.g., arid and humid basins), as well as different physiographic settings (e.g., flat and steep basins).

More broadly, we advocate for developing and carrying out analyses for SOMs tailored to other dissolved and suspended constituents, as distinct loop patterns are likely to emerge for different parameters. For instance, dissolved solids often exhibit dilution during hydrologic events, resulting in loops with a negative overall slope—a pattern rarely seen in sediment transport. To better capture these unique characteristics and understand their controls, we recommend creating C-Q SOM maps for other water quality parameters, such as nutrients, organic carbon, metals, and more.

**6 Conclusions**

This study introduces a novel method for characterizing C-Q hysteresis loops using the Self-Organizing Map (SOM) algorithm. Unlike traditional classification schemes that rely on nominal categories and fail to account for the gradual variability among loop types, the SOM-based framework captures the full continuum of hysteresis loop variability. Furthermore, it addresses limitations of hysteresis indices by providing a two-dimensional representation that captures greater detail in loop shapes,

offering a more nuanced depiction of the diversity within hysteresis loop types.





Our proof-of-concept application of the SOM algorithm for turbidity-discharge data demonstrates its efficacy for capturing key features of sediment hysteresis loops. This enabled the development of a *General T-Q SOM*, which captures the primary loop types associated with sediment transport hysteresis patterns. This map characterizes variations in rotational direction, loop amplitude, and, unlike traditional hysteresis indices, loop concavity. The inclusion of concavity adds a critical dimension

to hysteresis analysis, as it has been linked to the dominance of different flow pathways, reflecting, along with the rotational direction, the relative contributions of different water and sediment sources. Alternative T-Q SOMs could be developed to better capture loop types not represented by the *General T-Q SOM*, such as *Figure 8* loops or other uncommon patterns. We encourage use of the *General T-Q SOM* as a benchmark for future studies to facilitate discovery of novel hysteresis patterns. We demonstrate how the *General T-Q SOM* can be used to facilitate workflows for hysteresis analysis. For example, we show

how the map can be used to visualize hysteresis frequency distributions, facilitate hysteresis comparisons between watersheds, and link hydrologic processes to hysteresis patterns.

Finally, we discuss several promising avenues for application of the SOM algorithm for hysteresis analysis, including discerning seasonality of hysteresis patterns in individual watersheds, and, more broadly, applying the algorithm to characterize hysteresis behavior at the regional or continental scale. We advocate for the development of SOMs tailored to other dissolved

and suspended constituents, as distinct loop patterns are likely to emerge for different parameters.

## 7 Code and data availability

We have released a Python package (https://github.com/ArlexMR/HySOM), designed to facilitate sediment transport hysteresis analysis using the *General T-Q SOM*. This package provides access to the *General T-Q SOM* and the dataset used for its training, along with the SOM algorithm for training new SOM maps for additional constituents.

## 8 Author contribution

Marin-Ramirez: conceptualization, data curation, formal analysis, investigation, methodology, software, supervision, visualization, writing (original draft preparation)

Mahoney: conceptualization, investigation, funding acquisition, investigation, methodology, project administration, resources, software, supervision, validation, visualization, writing (original draft preparation)

McDaniel: conceptualization, data curation, formal analysis, writing (original draft preparation)

## 9 Competing interests

The authors declare that they have no conflict of interest.



## 10 Acknowledgements

We gratefully acknowledge the financial support of this research from the US Department of Transportation University
Transportation Centers Program Award #69A3552348335, National Science Foundation Award #2217685: *The Inclusive Mentoring Hub in Kentucky and West Virginia*, and National Science Foundation Award #2418789: *The Flooding in Appalachian Streams and Headwaters (FLASH) Initiative.* We would like to thank our collaborators at the University of Kentucky for allowing us to access the turbidity data collected nearby USGS gage 03289000 used in this study.

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
