# Peer review of "Technical Note: Analysis of concentration-discharge hysteresis loops using Self-Organizing Maps"

_EGUsphere, 2025_

## Author Comment (AC1)

**TECHNICAL NOTE: ANALYSIS OF CONCENTRATION-DISCHARGE HYSTERESIS LOOPS USING SELF-ORGANIZING MAPS**

**AUTHORS' RESPONSE TO REFEREE 1**

Responses in red

Overall, this manuscript is well written and supported with literature. You demonstrate strong scientific rigor, present informative figures, and provide a supportive narrative.

First and foremost, we sincerely appreciate your positive assessment of our manuscript. Your comments were both encouraging and helpful. We have carefully addressed your suggestions in the revised version, which we believe has improved the overall clarity and quality of the work.

Line 26: You mention that event-scale concentration has been employed for decades, but the oldest citation is only 5 years old (*Malutta et al., 2020*). Can you add some older/original literature in this first sentence to support your claim? Perhaps *Williams (1989), Hamshaw et al. (2018), Bettel et al. (2025)* since you mention in Line 225 that hysteresis loops were first recognized in these articles.

The citations originally included in this line are review papers that provide comprehensive discussions on concentration–discharge relationships, including its historical development. However, we agree it is valuable to cite some of the original, foundational studies to better support our claim. Accordingly, we will update the manuscript to include the following references:

*Heidel, S. G. (1956). The progressive lag of sediment concentration with flood waves. Eos, Transactions American Geophysical Union, 37(1), 56–66.*

*Williams, G. P. (1989). Sediment concentration versus water discharge during single hydrologic events in rivers. Journal of Hydrology, 111(1), 89–106. https://doi.org/10.1016/0022-1694(89)90254-0*

*Evans, C., & Davies, T. D. (1998). Causes of concentration/discharge hysteresis and its potential as a tool for analysis of episode hydrochemistry. Water Resources Research, 34(1), 129–137. https://doi.org/10.1029/97WR01881*

Section 2.2 – 2.4: I think these sections would be easier visualized with a workflow diagram that links to Figure 3. You could include a visualization for the process of training the model and finding the BMU. In the same workflow diagram, you can include a visualization for the process of using the DTW. Then, those can have an arrow pointing to the "SOM training" in Figure 3. Additionally, if possible, including the topological preservation and quantization accuracy into the diagram would help create a complete "picture" of the process. Although these three sections are written well, it is hard to visualize the order of the process. Additionally, Figure 3 in its current state is too general to provide a specific picture of the training process.

We agree that adding workflow diagrams will significantly enhance the clarity and visual structure of Sections 2.2 to 2.4. Accordingly, we have created three new diagrams to illustrate:

- The intuition behind the Dynamic Time Warping (DTW) algorithm and its distinction from Euclidean distance (Fig. 1).

- The workflow for training an individual Self-Organizing Map (SOM) (Fig. 2).
- The process for tuning SOM hyperparameters for hysteresis analysis, highlighting the role of quantization and topographic errors (Fig. 3).

Because DTW and SOM training algorithms are well-established and thoroughly described in the literature, we have opted to include Figures 1 and 2 in the Supplementary Material. In contrast, Figure 3 represents a methodological contribution tailored specifically to our use of SOMs for hysteresis analysis and will therefore be included in the main manuscript.

[Figure]

Figure 1. Comparison of Euclidean and Dynamic Time Warping (DTW) distances. The figure illustrates how DTW's flexible alignment yields smaller distance values than Euclidean distance when sequences (loops) follow similar paths but are temporally misaligned.

[Figure]

Start

**Inputs**
Map Size: $w, h$
Hyperparameters: $\alpha_0, \alpha_f, \sigma_0, \sigma_f$
Number of iterations: $N$
Sequence of loops: $S$

Set initial prototypes $\boldsymbol{m}^{\eta_k}$ for each
Node $\eta_k, k = 1, 2, \ldots, wh$
sampling randomly from $S$

Draw a random sample $\boldsymbol{X}$ from $S$

Identify the $BMU$ as the prototype with the
minimum distance to the sample $\boldsymbol{X}$

$$BMU = \underset{\eta_k}{\arg\min}\, DTW\left(\boldsymbol{X}, \boldsymbol{m}^{\eta_k}\right)$$

Apply the *decay function* to compute
learning rate $\alpha$ and neighborhood radius $\sigma$
for the current iteration $t$ :

$$\alpha = \alpha_0 \left(\frac{\alpha_f}{\alpha_0}\right)^{\frac{t}{N}} \qquad \sigma = \sigma_0 \left(\frac{\sigma_f}{\sigma_0}\right)^{\frac{t}{N}}$$

Compute the *neighborhood function* values
for each node, centered at the $BMU$:

$$h_{BMU,\eta_k} = \exp\left(-\frac{\left\lVert \boldsymbol{r}_{BMU} - \boldsymbol{r}_{\eta_k} \right\rVert^2}{2\sigma^2}\right)$$

$\boldsymbol{r}_{BMU}, \boldsymbol{r}_{\eta_k}$ are the map coordinates of the
BMU and node $\eta_k$, respectively

Update prototypes based on how close they
are to the BMU:

$$m^{\eta_k} \leftarrow m^{\eta_k} + \alpha \cdot h_{BMU,\eta_k} \cdot \left(\boldsymbol{X} - \boldsymbol{m}^{\eta\,k}\right)$$

Reached number of iterations?
$t \leq N$?

No

Yes

End

Figure 2. Workflow diagram illustrating the training process of a Self-Organizing Map. Parameters $w$ and $h$ represent the number of nodes along the horizontal and vertical dimensions. The learning rate $\alpha$, starts at $\alpha_0$ and gradually decreases to $\alpha_f$ following the decay function. Similarly, the neighborhood radius $\sigma$ evolves from $\sigma_0$ to $\sigma_f$. All remaining symbols are defined within the diagram.

[Figure]

Figure 3. Workflow for tuning SOM hyperparameters. A grid search is used to train multiple SOMs with varying hyperparameter combinations. The resulting maps are then assessed to identify the optimal model. First, the best SOM dimensions are determined by locating the point beyond which increases in size yield no significant improvement in quantization error (elbow method). Next, Pareto-optimal SOMs with the selected dimensions are visually inspected to select the map with the most coherent distribution of loop types.

Line 228: I recommend simply adding a parenthesis such as (as seen on the left rows in Figure 4) after the sentence to immediately direct your viewers eyes to the single-line, clockwise, and counterclockwise.

We will modify the sentence as follows:
*However, our analysis suggests that sediment hysteresis loop diversity can largely be explained by 17 loop types, encompassing different variations of single-line (first row in Fig. 4), clockwise (second row in Fig. 4), and counterclockwise (third row in Fig. 4) topologies.*

Figure 4 caption: I would also recommend adding additional information on the single-line, Figure 8, clockwise, and counterclockwise topologies. You have a lot of detail in Figure 1-3, and I would suggest continuing that format.

We will modify the caption as follows:

*Figure 4. Loop types considered in the training dataset. Except for Figure-L loops—introduced by Hamshaw et al. (2018)—all loop types were originally described by Williams (1989). We gathered 20 samples per loop type from 37 stream gages operated by the USGS across the United States. While past studies often lump together all subclasses within the broader categories of Single Line, Clockwise, and Counterclockwise loops, our dataset introduces a more granular classification that captures multiple shaper variations within each type, such as differences in concavity, which reflect the relative spread of the sedigraph with respect to the hydrograph, or Figure-L loops, which indicates strong decoupling between discharge and sediment concentration. Readers interested in the hydrological interpretation of these loop types are encouraged to consult the original studies.*

Line 256: It seems like Line 256 should be appended to line 255.

We agree.

Line 259: Would the highly distorted maps be defined by topological error (e.g. referring to section 2.3) and not the "topographic" error that is listed? If so, topographic is used throughout the paper (Line 169 and Line 325, Line 327, etc.) and would need corrected.

The terms *topological preservation* and *topographic error* refer to related but distinct concepts. Topological preservation describes the abstract ability of a SOM to maintain relative distances between data points when projecting them from high-dimensional space to a two dimensional map. Topographic error, by contrast, is a specific metric used to quantify the degree of topological preservation achieved by a trained SOM. This distinction is explained in section 2.3:

> ***Topological preservation*** *ensures that the map maintains the neighborhood relationships of the input space. Hence, a good topological preservation indicates that similar prototypes are placed close to each other and dissimilar prototypes are placed farther apart.* ***Topological preservation can be quantified using the topographic error***.

Therefore, the correct term in this context is topographic error. However, we mistakenly used the term "topological error" on Line 157, and we have corrected this in the revised version.

Line 262: A citation should be provided for the Pareto-optimal analysis.

The following reference will be added where the Pareto-optimal analysis is thoroughly described

Koppa, A.; Gebremichael, M.; Yeh, W. W.-G. Multivariate Calibration of Large Scale Hydrologic Models: The Necessity and Value of a Pareto Optimal Approach. *Advances in Water Resources* **2019**, *130*, 129–146. https://doi.org/10.1016/j.advwatres.2019.06.005.

Figure 8 caption: Please provide an explanation of what is shown in (a), (b), (c), (d), specifically.

We will modify the caption as follows:

Figure 8. Mapping of all samples from the curated loop dataset (see Fig. 3) to their respective Best Matching Units (BMUs) on the Kohonen map. For visual clarity, loop types and their mappings are organized into four distinct panels: (a) single-line loops; (b) clockwise loops including Async, Sync, Concave, and Figure-L; (c) counterclockwise loops including Async, Sync, Concave, and Figure-L; and (d) Figure-8 and Line+ loops. Acronyms used in the figure include: SP – Sync-Peak, AP – Async-Peak, CU – Concave-Up, CD – Concave-Down, ccw – Counterclockwise, cw – Clockwise, fL – Figure-L, f8 – Figure 8.

---

## Author Comment (AC3)

**TECHNICAL NOTE: ANALYSIS OF CONCENTRATION-DISCHARGE HYSTERESIS LOOPS USING SELF-ORGANIZING MAPS**

**AUTHORS' RESPONSE TO REFEREE 3**

**Responses in red**

The study by Ramirez et al "Analysis of concentration-discharge hysteresis loops using Self-Organizing Maps" describes a novel method to analyse hysteresis patterns in high-frequency concentration-discharge data.

The paper is generally very well written and I particularly like the innovative analyses and combination of methods to characterize and assess hysteresis loops and to assign event characteristics and catchment properties using an unsupervised machine learning algorithm. Also the figures are very nicely presented.

The presented method can provide a major step forward in the field of hysteresis analysis. However, despite having worked with hysteresis analysis in the hydrological and water quality context extensively, I have major difficulties understanding the content. I acknowledge that this is due to the fact that I have no background on SOM and Kohonen's algorithm, but given that this study is going to be published in HESS, I think a large part of the audience is likely to have a background in watershed hydrology. Therefore, I stress that the authors should significantly improve on the explanation of the methods and results. I hope the authors find my comments below useful in improving the manuscript.

We appreciate your positive comments and constructive feedback. Based on your suggestions, we plan to make several revisions to improve the clarity of the methods and results, especially for readers without prior experience using the SOM algorithm. Specifically, we will rework and expand section 2 and 3 of the manuscript to better explain the SOM method within the context of hysteresis analysis. We've prepared two additional figures to explain our workflow and will also provide additional clarification of the SOM algorithm herein and in the updated manuscript. We have attached our proposed edits to sections 2 and 3 at the end of this response (Appendix A.3). We believe that the proposed changes will make the manuscript much more accessible to the broader hydrology community. We expand upon our proposed changes to the manuscript below.

**General and major comments:**

The general workflow is hard to grasp and it is unclear what your python package can accomplish and what the user needs to do get all this to work - particularly because this is meant to be a technical note. Therefore, I think the paper would benefit from a figure/flowchart explaining the required steps and purpose of the steps from downloading/acquiring a dataset, via the curation of the dataset, training, refinement, application, to the resulting map. This would be like a summary map linking and expanding on figures 3, 4 and 5. It might make sense to move the current chapter 2.5 to the beginning of section 2 and expand with the points I highlighted.

Thank you for this detailed and helpful comment. We propose to restructure Sections 2 and 3 to improve clarity and accessibility of the manuscript (as shown in Appendix A.3 at the end of this response), especially for readers unfamiliar with SOM. Specifically, we propose the following:

- 1. Fig. 3 will be expanded (see Fig. 3 in appendix A.1) to show a general workflow for 1) generating a C-Q SOM and 2) analyzing a C-Q dataset with the generated C-Q SOM. The intent of this figure is to show what inputs are required by users, the outputs after training, and the types of analyses that can be conducted with the trained SOM.
- 2. We believe it may be too cumbersome for readers if we were to fully include the necessary information regarding the training and refinement of the SOM for hysteresis analysis in Fig. 3. Thus, we propose to add a new figure (see Fig. 4 in appendix A.1) to explain the SOM training algorithm and clarify key concepts like 'prototype,' 'BMU,' 'samples,' 'n-length sequence,' 'number of nodes,' 'random samples,' 'distance function,' etc.
- 3. Similarly, we propose to add another figure (see Fig. 5 in appendix A.1) to illustrate the process of hyperparameter tuning, including the elbow method and Pareto analysis. Together, Figs. 3, 4, and 5 will help to better show the training and refinement of the SOM for hysteresis analysis. We note that Figs. 3, 4, and 5 are meant to be agnostic of water quality parameters, therefore we will reserve information regarding curation of the dataset for sediment hysteresis analysis in Section 3.
- 4. In addition to incorporating the updated Fig. 3 and new Figs. 4 and 5, we propose to rework sections 2 and adjust section 3 to more clearly describe the proposed workflow in alignment with the added figures. Specifically, we will relocate the original Section 2.5 and Fig. 3 to the beginning of Section 2, as recommended. We will also introduce a new subsection that explains how a hysteresis loop is represented within the SOM framework, clarifying concepts such as "sample," "n-length sequence," and other terms that the reviewer found confusing (see more on this in the next response). Finally, we will revise the text to ensure that the workflow is clearly articulated with the updated figures (Appendix A.3).
- 5. We will also add text to Section 7 (Code and data availability) to provide additional information regarding the functionality, inputs, and outputs of the *HySOM* python package. The package has two primary sets of functions: 1) those that facilitate sediment transport hysteresis analysis using the *General T-Q SOM* and 2) those that can be used to train new SOMs for other constituents. Since the main goal of the manuscript is to describe the workflow for training an SOM for hysteresis loops and evaluating the efficacy of the algorithm for hysteresis analyses, detailed documentation on use of the package is provided separately in our GitHub repository.

For someone not familiar with SOM, I feel section 2 is written very abstract and is of very limited usefulness. First, I think it is extremely important that you stick to one definition and don't use a different word to describe the same SOM property. Perhaps this is the case, but I am doubtful, e.g. is a prototype and a sample the same (this is unclear in l.108)? Second, it would help if you could link the SOM properties (such as 'prototype', 'BMU', 'samples', 'n-length sequence', 'number of nodes', 'random samples', 'distance function' (distance between what?), 'topological preservation', 'quantization accuracy', 'topographic error', 'radius of influence') to actual properties of the C-Q data

analysis. I assume that the SOM algorithm properties you mention must be associated to some kind of 'metrics' that are derived from the C-Q data (such as duration, time steps, magnitude, difference, event and hysteresis properties). In my opinion it would improve the understanding of the methods tremendously if you could highlight such links wherever possible. For instance, would the number of nodes be similar to the 'sensitivity' of defining individual C-Q-events or to the number of different hysteresis classes, or...?

We thank the reviewer for these suggestions. As mentioned, we propose reworking section 2 and adjusting section 3 with the goal of better describing the SOM algorithm in the context of hysteresis analysis. Specifically:

- 1. We will revise the manuscript to ensure consistency when describing the components of the SOM algorithm. Furthermore, we will ensure that all SOM terminology is placed in the context of hysteresis. For convenience, we have defined several of these terms in this response and will ensure that each is clearly explained in the revised manuscript. An input sample is the data for a (measured) hysteresis loop, represented as a sequence of paired discharge and concentration values. A prototype is a hysteresis loop belonging to a node in the trained SOM that represents a cluster of similar samples, derived from the SOM algorithm. In other words, the prototype can be used to represent a subset of similar hysteresis loops in a dataset. There is a nuanced difference between prototype and BMU in that a prototype is a characteristic representation of C-Q hysteresis loops, while the BMU is the specific node in the SOM whose prototype best matches an individual storm event's hysteresis data. n-length sequence is a sequence of n C-Q pairs representing a given hysteresis loop (we recommend resampling the loops such that each event and prototype have a consistent dimension). Number of nodes refers to the size of the SOM, i.e., the number of distinct loop types needed to represent the hysteresis dataset. Random samples refers to a C-Q hysteresis loop that is randomly selected from the training dataset. Distance function refers to the measure of similarity (or dissimilarity) between two hysteresis loops (such as hysteresis data from a watershed and a prototype on an SOM). Quantization error measures how closely the hysteresis prototypes approximate the input hysteresis samples. Topographic error measures how well the SOM maintains neighborhood relationships from the input space, indicating whether similar prototypes are positioned near each other on the map. Finally, neighborhood radius refers to a hyperparameter which controls the transition between hysteresis prototypes. For example, a narrower radius may improve the quantization error of prototypes, but risks topological fragmentation, which can result in dissimilar hysteresis loops being placed close together.
  - These concepts will also be explained with the addition of the new Figs. 4 and 5 and their accompanying explanatory text.
- 2. We also propose to add a new section to the Supplementary Information (see Appendix A.2) that explains the distance function used in our SOM implementation (termed *Dynamic Time Warping, DTW*), which quantifies the difference between hysteresis data and prototypes. This section will describe how DTW operates in the context of hysteresis loops and its role in measuring loop similarity.
- 3. Note that there are nuanced differences between the terms topological preservation and topographical error, where the latter is one method (of several) to quantify the former. To reduce potential confusion from readers, we will only use the term topographic error in the revised

manuscript since this is what we're calculating to evaluate the degree to which the SOM maintains neighborhood relationships.

**Specific comments:**

l. 20: "while preserving the continuum of loop variability lost in classification schemes" this is unclear to me and could be explained in a little more detail here.

Thank you for pointing this out. We'll revise the sentence in the abstract to read: "...while preserving the gradual transitions between loop types that are often lost in classification schemes," which we believe better conveys the intended meaning. Further explanation will also be provided in the Introduction.

l. 138: would it be possible to link/explain C-Q data characteristics with the (some) terms of equation 1 (basically similar to my second major comment above)?

Equation 1 represents the SOM learning rule, where each hysteresis loop is denoted as X and the prototypes are denoted by  $m_t^i$  (  $m^{\eta_k}$  in the revised notation as shown in A.1 - Fig.4). Both X and  $m_t^i$  are  $n \times 2$  matrices representing a sequence of paired discharge and concentration values. This representation of hysteresis loops will be better explained in section 2 as follows: "For C-Q hysteresis analysis, we propose representing each loop as a sequence of paired discharge and concentration values, forming a matrix with dimensions  $n \times 2$ , where n indicates the number of data pairs used to represent the loop." Also, the factors in equation (1) are now better explained in the proposed Fig. 4 which details the training process.

Additionally, we will provide some more context for how components of SOM can directly be linked to hysteresis. Specifically, we will include the following text describing Eq. 1. "Finally, the prototype  $m^{\eta_k}$  of each node  $\eta_k$  is updated using the learning rule (Eq. 1), which adjusts the prototypes toward the input sample. In other words, each prototype hysteresis loop is iteratively refined to resemble C-Q hysteresis loops in the training data. This learning rule ensures that both the BMU and its neighboring nodes are moved closer to the current sample, allowing nearby prototypes to become more similar to the input loop, which preserves the continuity of patterns across the map."

l. 146-148 ff: between what exactly are these quantization errors calculated? It means you have to define a 'true' classification manually and map the SOM against this 'subjective' classification that compromizes your aim of "using SOM to discriminate and characterize loop types commonly seen in sediment transport literature"?

As defined in section 2.3 "[quantization error] is defined as the average distance between each sample and its BMU". Hence, calculating quantization error doesn't rely on any manual or subjective classification since this is an unsupervised algorithm. We believe the proposed modifications to the manuscript and new figures will make this much clearer. Particularly, we'll now define quantization error as follows "[quantization error] measures how closely the hysteresis prototypes approximate the input hysteresis samples. It is defined here as the average DTW distance between each sample and its BMU."

l. 175-176: similarity between two samples - is one sample Q and the other sample C?

Each sample refers to a full hysteresis loop, i.e., a matrix of Q-C data and not Q or C individually. We'll make this explicit at the start of Section 2 with the following text: "Each of these nodes act as a prototype, representing a cluster of similar samples (i.e., a number of individual hysteresis loops) in the training dataset." This will be further clarified by adding the following subsection:

**"2.3 Representing hysteresis loops for SOM**

In Fig. 3, samples (the light green shapes) and prototypes (the dark green shapes) are depicted as geometric shapes for illustrative purposes. However, the SOM algorithm requires that input data be represented as numeric arrays. In conventional SOM applications, each sample is represented as an n-dimensional vector and each resulting prototype is a vector of the same dimension. For C-Q hysteresis analysis, we propose representing each loop as a sequence of paired discharge and concentration values, forming a matrix with dimensions  $n \times 2$ , where n indicates the number of data pairs used to represent the loop. n therefore may be equal to the number of entries in the C-Q time series, however because hydrologic events usually vary in duration, we recommend resampling the data such that hysteresis samples and prototypes each share a consistent dimension. We expand on this more in section 3.

Alternative formats for samples and prototypes are also possible. For instance, Hamshaw et al. (2018) encoded loops as greyscale images with a resolution of  $28 \times 28$  pixels for use in a classification algorithm. However, in our implementation, representing loops as sequences of (Q, C) pairs yielded better results through the SOM algorithm."

l. 191ff: the difference between the first (training on a curated dataset?) and second (application to 'any dataset'?) phase of the SOM algorithm is not clear to me. Does it mean you need to split time series at one location/gauge into training and application? Or can you train at one location and then apply it to another location? Please explain the requirements and limitations in a bit more detail.

The first phase of our workflow (shown in Fig. 3) involves the actual training and creation of the SOM for a given constituent. The end goal of phase 1 is to create an SOM that should be generalizable for the constituent of interest, as explained in former section 2.5: "The first phase involves training an SOM to represent the spectrum of loop types for a particular dissolved or particulate constituent. To ensure the trained SOM adequately captures all primary loop types for a constituent, we recommend curating a dataset containing samples of all known loop types for the constituent under analysis during the training phase."

Hence, the dataset used to train the SOM should ideally capture the *full range of loop types* for a given constituent, which enables broader application of the SOM without the need for retraining. In our proof-of-concept we developed (and released through our Python package) an SOM trained on Turbidity-Discharge data, which, according to the existing sediment transport literature, generally captures the known shapes of sediment hysteresis. This information will be retained in the revised manuscript.

Importantly, we needed to verify that the SOM algorithm indeed could 1) replicate known sediment hysteresis shapes, and 2) assign loops to the correct prototypes. Our validation of the algorithm, as shown in former Fig. 8, which was facilitated *by the curated dataset*, gives us confidence that the SOM algorithm can indeed be used for sediment hysteresis analysis. So - to your question - training

of the SOM was based on data from many locations, and because the training dataset includes most of the known hysteresis loop types (barring irregular loops), the *General T-Q SOM* can be used to classify sediment transport loops from any watershed, without retraining. Of course, there may be cases where a watershed exhibits loop types that deviate from the patterns included in the *General T-Q SOM*. In that case, retraining our *General T-Q SOM* to include that loop type, or alternatively training a specific SOM for such a watershed may be required. There are certainly trade-offs for training an SOM for only one catchment, so we propose to make this explicit in the revised discussion by adding the following text to the discussion:

"Similarly, training an SOM on loops from a single watershed may reveal site-specific dynamics and deviations from the General T–Q SOM, offering valuable insights into local hysteresis behavior. However, such watershed-specific SOMs limit comparability across sites, as their prototypes are tailored to a single watershed. Additionally, if the number of available samples is small, the SOM may not be adequately trained, reducing its ability to capture meaningful structure in the hysteresis data."

The second phase of our workflow involves using the SOM derived from phase one to classify loop types in new datasets, as explained in former section 2.5. For example, one might input a dataset consisting of a sequence of T-Q loops, and the algorithm would then identify the Best Matching Unit for each hysteresis loop in the dataset. Therefore, one could use our *General T-Q SOM* to create a frequency distribution for their own watershed. A limitation of this, however, is that rare loop types may not be well captured on the map. In which case, as we aforementioned, one could either retrain our *General T-Q SOM* or train a new map (via phase 1 of the workflow). We discuss strategies to incorporate "less-common loop patterns" in Section 5.2 (Future use of the SOM algorithm for C-Q analyses).

l. l.194-201: 'curating a dataset with all known loop types' - this sounds to me like a major limitation of the method. Does it mean that you first need to analyse C-Q time series for 'old-style' loop types? Then, if C-Q relationships are very homogeneous in a catchment, it might be impossible to have a time series with different loop types - is the method then not applicable? For instance, different watersheds can cause quite different loop types for the same constituents - this limits the transferability of the method?

We would like to respectfully disagree with the statement that, 'curating a dataset with all known loop types,' is a major limitation of the method. The curated dataset serves three key purposes: (i) to represent the known hysteresis loops for a given constituent, thereby ensuring broad applicability of the resulting SOM; (ii) to guarantee representation of less frequent patterns; and (iii) to provide a reference for evaluating the SOM's ability to distinguish loops in line with conceptual classifications. We viewed the third point as especially important for this technical note because SOM has never been applied to water quality hysteresis loops (to our knowledge) and therefore its efficacy for characterizing hysteresis loops needed to be tested. The curated dataset facilitates this process, because we can see broadly what types of loops we included in the training dataset, and thus evaluate how well the *General T-Q SOM* could replicate those.

Our process to curate the dataset with all known loop types is detailed in section 3.1.1. Importantly, the curated dataset consisted of many loop types *across many catchments* to evenly represent the currently known sediment hysteresis loop types. This helped to mitigate potential bias in the training

dataset, since for example, we found that infrequent loops weren't well-captured in the SOM when we tried training the SOM for an individual watershed. Since curating the training dataset ensured even representation of less frequent patterns, we argue that our *General T-Q SOM* can be applied to all types of watersheds, including those that might have more homogenous C-Q relationships, as well as those with quite heterogeneous C-Q relationships.

With this being said, we do acknowledge that curating a dataset does have the limitation that rare loop types may not readily be represented by prototypes on the map. In a subsequent comment, you mention correctly that another approach for training an SOM for a given constituent would involve compiling a large dataset with a "sufficiently long / high number of time series, which implies that all possible loop types exist therein." This perhaps is one way to overcome limitations related to identification of rare loop types (and we hope that researchers will test this in the future). However, this method, too, has limitations. Specifically, compiling such a dataset would perhaps be an even larger (and longer) undertaking than what we have done in this manuscript. Second – we found that even with the curated dataset, the SOM algorithm occasionally has difficultly with representing loop types such as figure eight loops (this is a point that we mention in the discussion). Thus, if a loop occurs infrequently, or is of an increasingly complex nature, it is possible that the algorithm will have difficulty representing it on the map, which would remain unnoticed with the uncurated dataset. Finally, we should mention that it is possible to identify rare loop types via the General T-Q SOM. As we mention in the discussion, the quantization error can be used to "flag" loops that deviate highly from the loops represented on the General T-Q SOM, thereby serving as means to identify complex and rare loop types.

Regarding your question, "if C-Q relationships are very homogeneous in a catchment, it might be impossible to have a time series with different loop types - is the method then not applicable?" You are correct, though this applies primarily when training an SOM on a single catchment. In that case, the algorithm may only be exposed to a limited range of loop types, reducing its ability to generalize to other catchments. This limitation was one of the key reasons we opted to curate the dataset. That said, there is potential value in comparing individually trained SOMs across catchments, and we encourage future research to explore this direction. We'll make this explicit in the revised discussion by including:

"Similarly, training an SOM on loops from a single watershed may reveal site-specific dynamics and deviations from the General T–Q SOM, offering valuable insights into local hysteresis behavior. However, such watershed-specific SOMs limit comparability across sites, as their prototypes are tailored to a single watershed. Additionally, if the number of available samples is small, the SOM may not be adequately trained, reducing its ability to capture meaningful structure in the hysteresis data."

We really appreciate the reviewer for bringing up these very valid points.

l. 210: This figure is very (!) useful and I would strongly suggest to: (1) expand it to explain additional properties of the SOM algorithm that you introduced earlier and link it to the actual data properties, (2) refer to the figure earlier in the methods.

Thank you. We have accepted your suggestions as explained previously.

l. 233: compiling the 'curated dataset' through manual delineation looks like a major effort. First, you need to identify relevant events in the time series, then you need to derive loops and classify them... These steps usually require many subjective decisions (when is an event an event, which class types to use, which class is the resulting hysteresis loop in). How did you do this?

Regarding your statement, "First, you need to identify relevant events in the time series, then you need to derive loops and classify them." Yes, it is necessary to first identify the relevant events (which concurrently derives the hysteresis loops). This process would be requisite for anyone doing hysteresis analysis.

Classification of hysteresis loops is technically not required by the SOM algorithm, but again, we did this because we wanted to ensure that the known hysteresis loops for a constituent were represented. We should acknowledge that while curation of such a dataset is time-intensive, the process usually is done only once per constituent. Once trained, the SOM can be reused across locations without retraining, and it is also unnecessary to manually classify the loops from new watersheds. We will state this explicitly in the revised section 2:

"While dataset curation is a time-intensive task, requiring the extraction of multiple hydrologic events (for example, we included 340 events in our proof-of-concept; see Section 3.1), often from several watersheds, it is typically performed only once for each constituent's SOM. Once trained, the SOM can be readily shared and reused by other researchers without the need for retraining. For instance, along with this Technical Note, we released a trained SOM for sediment transport hysteresis analysis—referred to as the General T–Q SOM—which is publicly available for use in sediment hysteresis analysis studies."

Furthermore, we'll expand Section 3.1.1 to clarify our event selection process and why subjective decisions of this workflow are a minor concern for our study. We propose to add the following text: "Event delineation was supported by a custom-built application designed to label hydrologic time series.

Watershed selection and event delineation followed the criteria outlined below:

- Only rainfall-dominated hydrologic events were included; heavily regulated rivers and snowmelt-dominated watersheds were excluded.
- Multi-peak events were split into single-peak events if turbidity receded fully before the onset of the subsequent peak.
- Irregular loops (such as complex loops) or loop shapes that did not conform to the typologies illustrated in Fig. 6 were excluded.
- We then manually classified the loop shape to one of the typologies illustrated in Fig. 6 according to visual comparison until an even number of event types was identified.

We should note that while manual classification during data curation can be somewhat subjective, this is a minor concern since the SOM is not constrained to enforce strict boundaries between loop types and does not see the labeled classes. For example, whether a narrow clockwise loop is labeled as a single line or a clockwise loop may be debatable—but including it in training allows the SOM to learn the gradual transition between these shapes, independent of the assigned label."

l.257: I thought the number of nodes would have to be already defined in the previous step by arranging the manually delineation hysteresis loops into the grid you show in Figure S2? Is Figure S2

an output of your SOM, is it required as an input/during training of SOM? Or is it purely for information purpose?

The number of nodes is a hyperparameter that is optimized during training. It is not predefined. We believe that this will be clarified with the addition of the proposed Fig. 5 (appendix A.1). The reference to former Figure S2 in this section will be removed as it follows from the creation of the *General T-Q SOM*. Only the reference to figure S1 will be retained here. Figure S2 will be referenced only in the discussion section as it does leverage the arrangement of prototypes to show the distribution of loops.

l. 258-259: some additional methods and variables are mentioned here, such as "elbow method" or "number of epochs" - you didn't mention it in 2.2 Training process section.

The concept of "epoch" will be clarified as follows: "Training was conducted over five epochs, meaning the entire dataset was presented to the algorithm five times." Also, the elbow method will be introduced in the proposed Figure 5, which outlines the full hyperparameter optimization process.

l. 262: to me, this sounds like you calibrate the SOM map to the 'manual' delineation of loop types you conducted during dataset curation. This does not sound like 'unsupervised learning' (see also l. 282) and the 'subjective' classification you critizised in the introduction, is driving the properties of the SOM?

SOM training is fully unsupervised as stated in the introduction and section 2.1. The manual labels are not used at any point by the algorithm during the training process or in the application. We created labels for each of the loops in the curated training dataset, however, this was merely to ensure that we were representing the known types of hysteresis loops in the training data. As mentioned, the curated dataset serves three key purposes: (i) to represent the known hysteresis loops for a given constituent, thereby ensuring broad applicability of the resulting SOM; (ii) to guarantee representation of less frequent patterns; and (iii) to provide a reference for evaluating the SOM's ability to distinguish loops in line with conceptual classifications.

While manual classification during data curation can be somewhat subjective, this is a minor concern since the SOM is not constrained to enforce strict boundaries between loop types and does not see the labels. For example, whether a narrow clockwise loop is labeled as a single line or a clockwise loop may be debatable—but including it in training allows the SOM to learn the gradual transition between these shapes, independent of the assigned label.

l.283-l.287 here you finally mention that the whole manual delineation/classification is only needed to accomplish the curation of the dataset. Why is this needed? Couldn't I simply take a sufficiently long / high number of time series which implies that all possible loop types exist therein? - ok later you mention that you don't want an uneven distribution of loop classes - but isn't forcing a similar distribution introducing a bias? What would happen if you don't use this dataset curation (perhaps this can be elaborated on in the discussion).

As stated before, the curated dataset serves three key purposes: (i) to represent the known hysteresis loops for a given constituent, thereby ensuring broad applicability of the resulting SOM; (ii) to guarantee representation of less frequent patterns; and (iii) to provide a reference for evaluating the

SOM's ability to distinguish loops in line with conceptual classifications. We will better highlight this in the reworked section 2:

"To ensure the trained SOM adequately captures the diversity of loop patterns, we recommend curating a training dataset that includes representative examples of all known loop types for the constituent under investigation. This step is essential not only for achieving comprehensive representation of known loop types, but also for evaluating the algorithm's ability to distinguish and characterize hysteresis patterns, as illustrated as the last step of the Training phase (Fig. 3). Moreover, a curated and balanced training dataset mitigates bias toward more frequently occurring loop types in watersheds by incorporating a uniform number of samples for each type."

Training an SOM on a large, uncurated dataset is certainly possible, and may be a good future use of SOM. However, for the reasons outlined above, we found dataset curation to be the most suitable approach for our study.

We propose to state this explicitly in the revised discussion:

"While we chose to curate a dataset containing representative sediment hysteresis patterns for training the General T-Q SOM, alternative uses of SOM are worth exploring. For example, training on large uncurated datasets can serve as a powerful exploratory tool, leveraging SOM's visualization strengths to uncover dominant or previously unnoticed loop patterns. Note, however, that less frequent patterns may not be properly captured in the trained SOM due to the learning algorithm's limited exposure to rare types during training (Douzas and Bacao, 2017)."

We should also mention that a balanced training dataset doesn't impose a distribution on the resulting SOM. While it imposes a (uniform) distribution on the training dataset, this is done to ensure that both common and uncommon loop types are represented during training. However, the actual frequency of loop types in any watershed is revealed later when mapping new data to the trained SOM. Curation also helps us evaluate the SOM's ability to distinguish known loop types, as explained. While training on a large, uncurated dataset is possible, we recommend curation so the trained SOM can be evaluated. However, we also acknowledge that training an SOM on large, uncurated datasets could provide additional insights.

l. 295 suggest to add data sources of the catchment characteristics and how these were derived/extracted.

We will add the following information in section 3.2:

"Watershed boundaries were obtained from the USGS StreamStats online application, watershed slope was calculated using the USGS 3DEP (10m) digital elevation model, urban area was extracted from the 2021 National Land Cover Database and soil texture was extracted from the Probabilistic Remapping of SSURGO (POLARIS) database (Chaney et al., 2019)."

l. 300-307: The different methods for the three watersheds are confusing. Why additional variables only for 03289000 and not for the others? Why Zuecco-indices for the two others and not for 03289000? Suggest to explain how 'old-water to event-water' was calculated.

We will clarify in Section 3.2 that the three watersheds were selected to illustrate two distinct types of analysis: (1) loop type frequency (gages 07364130 and 03254480), and (2) associations with hydrologic variables (gage 03289000):

"These watersheds were selected to illustrate two primary analyses for characterizing hysteresis patterns: (1) exploring the frequency distribution of loop types within a watershed (gages 07364130 and 03254480), and (2) identifying associations between loop types and hydrologic variables (gage 03289000)."

The additional variables for 03289000 come from a previous study (cited in the manuscript), and we chose not to detail their derivation here since it's already covered there and not central to this paper's focus. We also excluded hysteresis indices for 03289000 as they didn't add value to the analysis.

l. 329-333: why not selecting 0.02/0.7 which would seem to have a lower euclidean distance error than the one you chose.

The 0.02/0.7 SOM was indeed one of the Pareto-optimal maps we reviewed (see figure below). As explained in Section 3.1.2, we selected the final SOM based on visual evaluation, favoring smoother transitions and clearer separation between contrasting loop types. While both maps performed similarly in terms of QE and TE (both were along the pareto frontier), the chosen SOM offered a more intuitive layout for understanding loop variability.

Note for example how the transition from clockwise to counterclockwise loops follows a clear diagonal (lower-left to upper-right), while in the discarded SOM this transition is more scattered and harder to interpret. Both maps capture similar loop types, but we prioritized the one that offers a more intuitive layout for understanding loop variability.

We will add this figure and a short explanation to the SI so readers can better grasp the visual evaluation of Pareto-optimal SOMs.

l. 340ff: Given my difficulties with the methods, I am also confused by chapters 4.1.2 and 4.1.3. Chapter 4.1.2 is based on the curated dataset (=manual classification as far as I understand your earlier explanation) - but it also shows a 'trained SOM' (l.341). Then in 4.1.3, (l.370) you mention that "manual classification was not seen by the model as part of the training process" - This is confusing.

The General T–Q SOM was trained using the curated dataset, but as an unsupervised algorithm, it never sees or uses the manual labels during training. The curated dataset simply ensures a balanced representation of loop types and allows us to evaluate how well the SOM distinguishes between them. We believe that section 4 will be much easier to understand with the proposed figures and revisions to section 2.

**Minor comments:**

l. 74-79 the information given in the caption partly duplicates information given in the main text. Suggest to streamline this.

Thank you for the suggestion. We prefer self-explanatory figures for readers who may not refer to the main text.

l. 228: suggest to write "figure eight" to avoid confusion with Figure 8.

Accepted. We now use the term figure-eight.

l. 313: something is missing here "a two dimensional..." array? matrix?

The right word is *index*. Corrected.

1.400: reference and explain a, b, c and d in the caption.

**Accepted. The new caption will read:**

"Figure 10. Mapping of all samples from the curated loop dataset to their respective BMUs on the General T-Q SOM. For visual clarity, loop types and their mappings are organized into four distinct panels: (a) single-line loops; (b) clockwise loops including Async, Sync, Concave, and Figure-L; (c) counterclockwise loops including Async, Sync, Concave, and Figure-L; and (d) Figure-eight and Line+loops. Acronyms used in the figure include: SP – Sync-Peak, AP – Async-Peak, CU – Concave-Up, CD – Concave-Down, ccw – Counterclockwise, cw – Clockwise, fL – Figure-L, f8 – Figure-eight."

l.432: suggest not to name this technical note a "report"

**Accepted.**

l.475: the three arrows are hard to distinguish. It might make sense plotting them in three different colors and since they seem to overlap with a transparency value

Thanks for the suggestion. The arrows don't overlap. Antecedent discharge  $(Q_{5D})$  has a near-zero coefficient, so its arrow is simply too short to be visible. We'll make this clear in the text: "In contrast, antecedent discharge shows negligible association, reflected in a near-zero coefficient and short arrow."

**Appendix:**

**A.1 Proposed figures to better explain the workflow for using SOM in hysteresis analysis**

Figure 3. Workflow to generate an SOM for C-Q hysteresis loops (training phase) and apply the SOM for C-Q hysteresis analyses in watersheds (analysis phase). Here, we illustrate the generation of the SOM using different shapes, which are analogous to the hysteresis loop types that might be found for a dissolved or particulate constituent in a watershed. In the bottom panel (analysis phase), we demonstrate how hysteresis loops from a new dataset get mapped to the trained SOM, where the shade of orange represents the frequency with which the shape occurs in the dataset. The *HySOM* python package (see section 7) allows users to implement this workflow.

Figure 4. Workflow for training an SOM.  $\alpha$ : learning rate,  $\sigma$ : neighborhood radius, subscripts 0, f indicate initial (first iteration) and final (last iteration) values, respectively. Q: Discharge, C: Concentration, n: length of the sequence of (Q,C) data pairs representing a loop (section 2.3). Other symbols are defined in the figure.

Figure 5. Workflow for fine-tuning SOM hyperparameters using a combination of quantitative metrics and qualitative assessment. The process begins with training multiple SOMs across a grid of map sizes, learning rates, and neighborhood radii. Quantization error is evaluated to identify the optimal map size using the elbow method. A subset of high-quality maps—selected from the Pareto frontier of topographic and quantization errors—is then examined in detail. SOM selection is based on visual inspection, prioritizing maps that exhibit coherent transitions between similar loop types and clear separation between contrasting ones. Finally, retraining of the selected SOM may enhance quantization accuracy.

**A.2 Proposed new section S in the SI to better explain the DTW algorithm**

**S1. Dynamic Time Warping**

Figure S1 illustrates the difference between Euclidean distance and dynamic time warping (DTW) for one-dimensional sequences. Euclidean distance compares time series by matching values at the same time index, resulting in larger distances if sequences are misaligned. In contrast, Dynamic Time Warping (DTW) flexibly matches points across time to minimize overall distance. For example,  $x_3$  in the first sequence is matched with  $y_5$  in the second sequence.

Figure S1. Comparison between Euclidean distance and Dynamic Time Warping distance for onedimensional sequences

The same principle applies to two-dimensional sequences, as illustrated by the two hysteresis loops in Fig. S2. DTW flexibly matches nearby points along the trajectory on the Q-T plane, resulting in lower distances when the loops follow similar paths. In contrast, Euclidean distance is sensitive to time misalignments and compares points at fixed positions—leading to larger distances even when the overall loop shapes are similar.

Figure S2. Comparison between Euclidean distance and Dynamic Time Warping distance for two-dimensional sequences

**A.3 Proposed Revisions to Sections 2 and 3**

2 Self-Organizing Maps for C-Q hysteresis analysis

**2.1 Brief description of SOM**

The SOM algorithm (Kohonen, 1982; Kohonen, 1990) is an unsupervised learning technique widely applied in various fields, including clustering, classification, manifold learning, dimensionality reduction, and data visualization (Miljković, 2017). In water resources, SOM has been applied to diverse purposes such as rainfall-runoff modelling, regionalization, clustering of water quality data, analysis of land use and land cover, and more (Kalteh et al., 2008; Clark et al., 2020). This algorithm generates a discrete representation of a dataset known as a feature map or Kohonen map (Miljković, 2017), which typically takes the form of a two-dimensional grid of nodes arranged in either a rectangular or hexagonal lattice. Each of these nodes acts as a *prototype*, representing a cluster of similar *samples* (here, a number of individual hysteresis loops) in the training dataset. The spatial arrangement of prototypes on the map preserves the topological structure of the input training data such that similar prototypes are placed close to each other, while dissimilar prototypes are placed farther apart. In the context of hysteresis, the input samples include paired discharge and concentration values representing hysteresis events, and the output of the SOM algorithm is a set of prototype loops that represent characteristic C-Q behavior arranged in a continuum.

By projecting the spectrum of loops into a two-dimensional space, SOM retains more information compared to the hysteresis index—which offers only a one-dimensional projection—while maintaining a low dimensionality that enables easier analysis and visualization. For instance, the frequency distribution of loop types can be visualized as a heatmap or scatter plot mapped onto the trained SOM. Additionally, this map can serve as a tool to visualize and quantify the relationships between loop types and their potential hydrologic controls, as demonstrated in Section 4.

**2.2 Applying SOM in hysteresis analysis**

We propose using SOM for C-Q hysteresis analysis following the two-phase workflow illustrated in Fig. 3. The first phase—the *Training phase*—involves constructing an SOM to represent the spectrum of hysteresis loop types associated with a specific dissolved or particulate constituent. To ensure the trained SOM adequately captures the diversity of loop patterns, we recommend curating a training dataset that includes representative examples of all known loop types for the constituent under investigation. This step is essential not only for achieving comprehensive representation of known loop types, but also for evaluating the algorithm's ability to distinguish and characterize hysteresis patterns, as illustrated as the last step of the Training phase (Fig. 3). Moreover, a curated and balanced training dataset mitigates bias toward more frequently occurring loop types in watersheds by incorporating a uniform number of samples for each type.

While dataset curation is a time-intensive task, requiring the extraction of multiple hydrologic events (for example, we included 340 events in our proof-of-concept; see Section 3.1), often from several watersheds, it is typically performed only once for each constituent's SOM. Once trained, the SOM can be readily shared and reused by other researchers without the need for retraining. For instance, along with this Technical Note, we released a trained SOM for sediment transport hysteresis analysis—referred to as the General T–Q SOM—which is publicly available for use in sediment hysteresis analysis studies.

The output of the *Training phase* is a trained SOM composed of coherently arranged prototypes that reflect the range of loop types present in the training dataset. Further details on the training and evaluation procedures are provided in section 2.5.

The *Analysis phase* consists of using the trained SOM to classify loop types in new datasets, provided they correspond to the same constituent of interest. First, each loop in the dataset is mapped to its most similar loop prototype, known as the Best Matching Unit (BMU), based on distance function as described in section 2.4. This classification enables systematic analysis of hysteresis patterns across watersheds. For example, by quantifying the number of samples assigned to each prototype, researchers can characterize the frequency distribution of loop types and investigate their associations with hydrological variables, thereby providing valuable insights into the underlying controlling mechanisms.

Figure 3. Workflow to generate an SOM for C-Q hysteresis loops (training phase) and apply the SOM for C-Q hysteresis analyses in watersheds (analysis phase). Here, we illustrate the generation of the SOM using different shapes, which are analogous to the hysteresis loop types that might be found for a dissolved or particulate constituent in a watershed. In the bottom panel (analysis phase), we demonstrate how hysteresis loops from a new dataset get mapped to the trained SOM, where the shade of orange represents the frequency with which the shape occurs in the dataset. The *HySOM* python package (see section 7) allows users to implement this workflow.

**2.3 Representing hysteresis loops for SOM**

In Fig. 3, samples (the light green shapes) and prototypes (the dark green shapes) are depicted as geometric shapes for illustrative purposes. However, the SOM algorithm requires that input data be represented as numeric arrays. In conventional SOM applications, each sample is represented as an n-dimensional vector and each resulting prototype is a vector of the same dimension. For C-Q hysteresis analysis, we propose representing each loop as a sequence of paired discharge and

concentration values, forming a matrix with dimensions  $n \times 2$ , where n indicates the number of data pairs used to represent the loop. n therefore may be equal to the number of entries in the C-Q time series, however because hydrologic events usually vary in duration, we recommend resampling the data such that hysteresis samples and prototypes each share a consistent dimension. We expand on this more in section 3.

Alternative formats for samples and prototypes are also possible. For instance, Hamshaw et al. (2018) encoded loops as greyscale images with a resolution of  $28 \times 28$  pixels for use in a classification algorithm. However, in our implementation, representing loops as sequences of (Q, C) pairs yielded better results through the SOM algorithm.

**2.4 Quantifying similitude between samples: The distance function**

The distance function is a critical element of the SOM algorithm as it measures the similarity between a sample and each prototype. In the context of hysteresis, the distance function measures the similarity (or dissimilarity) between two hysteresis loops. This function directly affects the SOM algorithm's ability to extract the main patterns from the input data and arrange them in a coherent manner. Furthermore, this function is required during the Analysis phase (Fig. 3) to map new hysteresis data to their associated loop prototype (BMU). While Euclidean distance is the default selection in most SOM applications, any function that computes the degree of similarity between two numeric arrays can be used such as Cosine similarity, Manhattan distance, Minkowski distance, etc. (Samarasinghe, 2016). Given our representation of hysteresis loops as sequences of Q-C data pairs, we propose using Dynamic Time Warping (DTW) to measure similarity between loops.

DTW is an algorithm that measures the similarity between two time series. It has proven effective in clustering and classification across various domains involving sequential data (Ding et al., 2008). Its primary advantage lies in its ability to prioritize the overall shape of the temporal sequences over a strict match of individual data points comprising the sequence. This is achieved by non-linearly stretching or compressing the time axis of one sequence to achieve optimal alignment with another, prior to computing the Euclidean distance between the aligned sequences. The resulting DTW distance reflects this alignment: it is upper bounded by the Euclidean distance when no alignment is possible, but yields a lower value when warping improves the match, indicating greater shape similarity. A comparison between Euclidean and DTW distances is presented in Supplementary Information (section S1). Further details on the DTW algorithm and its application in water resources can be found in Lee et al. (2020) and Dupas et al. (2015), while its use with two-dimensional time series data is discussed in Shokoohi-Yekta et al. (2017).

**2.5 SOM training and evaluation**

**2.5.1 SOM training and hyperparameter optimization**

**Map initialization**

The algorithm for training an SOM for hysteresis analysis is illustrated in Fig 4. First, training hyperparameters must be set. This includes the number of nodes in the map lattice, which

determines the number of prototypes—and therefore, the number of distinct loop types—to be included in the trained map. While a larger map can represent a dataset with higher accuracy, they are often more difficult to visualize and interpret. Additionally, larger maps may produce prototypes that represent too few or none of the input samples, making the results more susceptible to noise and outliers. Ultimately, defining the map size relies on heuristic rules, domain intuition, and visual examination of the dataset (Vesanto, 2000; Kohonen, 2013). Importantly, however, the optimal size of the map should be refined iteratively, expanding the number of nodes until increases no longer result in meaningful improvements in the quality of the map (Céréghino and Park, 2009). We discuss metrics to assess the map quality in Section 2.5.2.

Additional hyperparameters include the initial and final learning rate and neighborhood radius  $(\alpha_0 \ \alpha_f, \sigma_0 \ \text{and} \ \sigma_f)$ . As explained below, they control how much the map prototypes are adjusted during each step of the training process. Finally, the number of iterations define the length of the training process.

Once the map size is defined, initial prototypes must be assigned to each node. Random initialization using samples from the training dataset is commonly used, although more elaborate initialization approaches can also be employed (Attik et al., 2005).

**Training loop**

With this initial, untrained map, the training process is conducted by sequentially feeding random samples (i.e., a C-Q hysteresis loop that is randomly selected from the training dataset) and adjusting the prototypes to better match the input data. For each sample (i.e., hysteresis loop) X, the BMU is identified as the node whose prototype exhibits the highest similarity to the input sample as measured by the DTW function. Once the BMU is identified, the learning parameters for the current iteration—learning rate ( $\alpha$ ) and neighborhood radius ( $\sigma$ )—are computed using a decay function. This decay function gradually reduces  $\alpha$  and  $\sigma$ , resulting in a transition during training from an initial ordering and placement phase where prototypes broadly align with the spatial structure of the input data, to a fine-tuning phase where prototypes are refined to better represent the input samples (Samarasinghe, 2016).

Common choices of decay functions include hyperbolic, exponential, and linear (Kohonen, 2013). In our implementation, we adopted an exponential decay rate which varies from initial to final values defined during the map initialization (Fig. 4).

In the next step of the training loop, neighborhood values  $(h_{BMU,\eta_k})$  are computed. These values control the extent to which the prototype of each node  $\eta_k$  is updated during the current iteration based on the Euclidean distance—in map coordinates—between  $\eta_k$  and the BMU. This function plays a critical role in enabling SOM to produce smooth transitions across neighboring prototypes (i.e., ensuring that similar hysteresis loops are placed near one another). The neighborhood value reaches its maximum at the BMU and decreases with increasing distance from it, with the rate of decay being controlled by  $\sigma$ . In our implementation, we used the Gaussian kernel (Fig. 4), which is the most commonly used neighborhood function in SOM applications (Kalteh et al., 2008).

Finally, the prototype  $m^{\eta_k}$  of each node  $\eta_k$  is updated using the learning rule (Eq. 1), which adjusts the prototypes toward the input sample. In other words, each prototype hysteresis loop is iteratively

refined to resemble C-Q hysteresis loops in the training data. This learning rule ensures that both the BMU and its neighboring nodes are moved closer to the current sample, allowing nearby prototypes to become more similar, which preserves the continuity of patterns across the map. The magnitude of adjustment decreases over time and with distance from the BMU, as determined by the decay and neighborhood functions.

$$m^{\eta_k} \leftarrow m^{\eta_k} + \alpha \cdot h_{BMU,\eta_k} \cdot (\mathbf{X} - \mathbf{m}^{\eta_k}) \tag{1}$$

Figure 4. Workflow for training an SOM.  $\alpha$ : learning rate,  $\sigma$ : neighborhood radius, subscripts 0, f indicate initial (first iteration) and final (last iteration) values, respectively. Q: Discharge, C: Concentration, n: length of the sequence of (Q,C) data pairs representing a loop (section 2.3). Other symbols are defined in the figure

**2.5.2 Hyperparameter optimization**

Training hyperparameters—such as map size, learning rate, and neighborhood radius—directly influences the SOM's ability to preserve data topology and accurately represent input samples. Therefore, optimal values should be tailored to the specific use case. To guide hyperparameter selection, we rely on two widely used SOM quality metrics: topographic error and quantization error. These metrics provide complementary insights into the structural fidelity and representational accuracy of the trained map.

Topographic error quantifies the degree to which the SOM maintains neighborhood relationships from the input space. A low topographic error indicates that similar prototypes are positioned close together, while dissimilar ones are placed farther apart. In the context of hysteresis, this means that loops with similar amplitude, rotational direction and shape should be placed close together. This metric is defined as the fraction of samples for which the Best Matching Unit (BMU) and the second BMU (i.e., the prototype with the second smallest distance to the input vector) are not adjacent on the map (Kiviluoto, 1996; Pölzlbauer, 2004). Lower values are preferred.

Quantization error, on the other hand, measures how closely the hysteresis prototypes approximate the input hysteresis samples. It is defined here as the average DTW distance between each sample and its BMU (Pölzlbauer, 2004). Lower quantization errors indicate better representational accuracy.

These two metrics often represent competing objectives. Reducing one metric may increase the other. Therefore, a balance must be achieved by evaluating SOMs trained under varying hyperparameter configurations. Trade-offs to consider include:

- Map size: Increasing the number of nodes generally reduces quantization error, as more
  prototypes can better approximate the dataset. However, larger maps may complicate
  visualization and interpretation. In the context of hysteresis analysis, the map size
  determines the number of distinct loop types (i.e., hysteresis loop prototypes) to be included
  in the trained map.
- Neighborhood radius: A wider radius promotes smoother transitions between hysteresis
  prototypes, reducing topographic error but potentially increasing quantization error.
  Conversely, a narrower radius sharpens prototype specialization, improving quantization
  error such that hysteresis loops are mapped to their prototypes relatively well, but risking
  topological fragmentation, which can result in dissimilar hysteresis loops being placed close
  together.
- Learning rate: Influences the speed and stability of convergence, with its decay profile affecting both error metrics.

In addition to quantitative metrics such as topographic and quantization errors, SOMs trained on hysteresis loops can be evaluated qualitatively as the prototype distribution can be easily visualized. The spatial distribution of prototypes offers a visual indication of topological preservation, with smooth and coherent transitions between loop types serving as a desirable trait. To fine-tune training hyperparameters, we propose the workflow illustrated in Fig. 5. First, multiple SOMs are trained across a grid of map sizes, learning rates, and neighborhood radii. The optimal map size is identified by examining the relationship between quantization error and number of nodes. While a decreasing trend is expected, the elbow method (Nainggolan et al., 2019) helps identify the point of diminishing returns. Once the optimal size is selected, a subset of high-quality maps—trained with varying learning rates and neighborhood spreads—is extracted for closer inspection. These maps are chosen from the Pareto frontier of topographic and quantization errors.

Visual assessment of these candidate maps is then performed, guided by conceptual understanding of loop type similarities. In this application, we prioritized maps that exhibited smooth transitions between similar loop types and clear separations between contrasting ones. Finally, in our proof-of-concept, retraining the selected SOM further improved map quality. While the initial selection emphasized topological arrangement, retraining enhanced quantization accuracy, yielding final error metrics that surpassed the original Pareto frontier.

Figure 5. Workflow for fine-tuning SOM hyperparameters using a combination of quantitative metrics and qualitative assessment. The process begins with training multiple SOMs across a grid of map sizes, learning rates, and neighborhood radii. Quantization error is evaluated to identify the optimal map size using the elbow method. A subset of high-quality maps—selected from the Pareto frontier of topographic and quantization errors—is then examined in detail. SOM selection is based on visual inspection, prioritizing maps that exhibit coherent transitions between similar loop types and clear separation between contrasting ones. Finally, retraining of the selected SOM may enhance quantization accuracy.

**2.6 Evaluation of a trained SOM**

The final step of the training phase (Fig. 3) involves evaluating the SOM's ability to represent the full spectrum of loop types included in the curated dataset. We propose visual inspection as the most suitable approach, as it allows researchers to assess whether the distribution of prototypes reflects the diversity of loop shapes in the curated dataset and whether their arrangement aligns with conceptual expectations of similarity and differences. For instance, in our proof-of-concept, we examined whether loops from closely related classes (e.g., async-peak and sync-peak; see Fig. 6)

were mapped to neighboring prototypes—indicating the SOM's ability to recognize shared geometric traits consistent with expert classifications.

This evaluation helps identify which loop features are effectively captured by the SOM and which may require more nuanced analysis. Crucially, because SOM training is fully unsupervised, it does not rely on manual labels; instead, it organizes loops based solely on shape similarity, as defined by the DTW distance function.

Although quantitative metrics such as confusion matrices could be used to assess classification performance, we chose not to apply them here. Such metrics would require converting the SOM output into discrete categories, undermining its key advantage: preserving smooth transitions and continuous variation between loop types.

**3 Applying SOM to turbidity-discharge hysteresis loops**

We employ turbidity-discharge (T-Q) data to develop a proof-of-concept and demonstrate the proposed workflow for training and applying SOM in C-Q hysteresis analysis. We stress, however, that this workflow is adaptable to other constituents such as nutrients, dissolved solids, metals, and more. Future studies are encouraged to explore its application across a broader range of constituents. The following sections detail the implementation of our proposed workflow (as shown in Fig. 3).

**3.1 Training phase**

**3.1.1 Data curation and preprocessing**

We surveyed the sediment transport literature to compile the primary loop types included in the curated dataset (Fig. 6). Our review indicates that the majority of sediment hysteresis loop types recognized in the literature were originally identified by Williams (1989), with the *Figure L* loop introduced by Hamshaw et al. (2018). Additional studies have introduced nuanced variations of these loop types; for example, Bettel et al. (2025) introduced the "J" loop for a karstic system in Kentucky, USA which resembles concave-up loops. However, our analysis suggests that sediment hysteresis loop diversity can largely be explained by the 17 loop types identified in Fig. 6, encompassing *single-line*, *Figure-eight*, *clockwise*, and *counterclockwise* topologies. Irregular loops were excluded due to their lack of a standardized classification system. While additional loop types may exist beyond those included, they are generally infrequent and relevant only to specific studies or watersheds. We discuss the incorporation of less common loops in Section 5 and encourage the development of tailored SOMs to accommodate specialized applications.

The curated dataset consists of 20 samples for each loop type, resulting in a total of 340 loops. Loops were manually delineated using publicly available discharge and turbidity data collected at 15-minute intervals by the USGS across 37 stream gauges in the United States. Data were retrieved from the National Water Information System using the *DataRetrieval* Python package (Hodson et al., 2023). Event delineation was supported by a custom-built application designed to label hydrologic time series.

Watershed selection and event delineation followed the criteria outlined below:

- Only rainfall-dominated hydrologic events were included; heavily regulated rivers and snowmelt-dominated watersheds were excluded.
- Multi-peak events were split into single-peak events if turbidity receded fully before the onset of the subsequent peak.
- Irregular loops (such as complex loops) or loop shapes that did not conform to the typologies illustrated in Fig. 6 were excluded.
- We then manually classified the loop shape to one of the typologies illustrated in Fig. 6
  according to visual comparison until an even number of event types was identified.

We should note that while manual classification during data curation can be somewhat subjective, this is a minor concern since the SOM is not constrained to enforce strict boundaries between loop types and does not see the assigned labeled. For example, whether a narrow clockwise loop is labeled as a *single line* or a *clockwise Sync-Peak* loop may be debatable—but including it in training allows the SOM to learn the gradual transition between these shapes, independent of the assigned label.

The Supplementary Information provides detailed site information and data periods (Table S1), while the complete set of loops is presented in Fig. S3.

Figure 6. Loop types considered in the training dataset. All loop types, with the exception of *Figure-L* loops, were initially described by Williams (1989). *Figure-L* loops were introduced by Hamshaw et al. (2018)

The SOM algorithm requires samples to be represented as arrays with an equal number of elements. Given that loops in the dataset consist of variable-length sequences of Q-T data pairs, we implemented a preprocessing procedure to convert all loops into sequences of equal length, ensuring compatibility with the SOM algorithm. This procedure is illustrated in Fig. 7 and involves the following steps. First, we applied a moving median with a 5-point window to the turbidity values to reduce random noise (Fig. 7b). Second, the Q-T data were scaled to a [0,1] interval using minmax normalization based on the minimum and maximum values for each event. This scaling, commonly applied prior to the calculation of hysteresis indices, facilitates comparison between loops of different magnitudes (Lloyd et al., 2016); and third, we interpolated the resulting variablelength, scaled T-Q data (Fig. 7c) to produce equal-length sequences (Fig. 7d) of 100 data pairs. The interpolation ensures that data points are equally spaced in the Q-T plane.

Figure 7. Workflow illustrating data preprocessing steps: a) original discharge and turbidity data. b) A 5-point moving median was applied to turbidity data to mitigate outliers. c) Q-T data were minmax normalized to a [0,1] interval. d) the scaled, variable-length data were interpolated into equallength sequences of 100 data pairs.

**3.1.2 SOM training and hyperparameter optimization**

SOM training and hyperparameter tuning followed the workflows explained in section 2.5. SOM sizes ranged from 5x5 to 13x13 nodes, with initial neighborhood radii varying from 0.5 to 13 and initial learning rates from 0.05 to 0.9. Final values for neighborhood radius and learning rate were set at 0.3 and 0.01, respectively. Training was conducted over five epochs, meaning the entire dataset was presented to the algorithm five times, resulting in 1700 iterations per hyperparameter combination (340 samples  $\times$  5 epochs). In total, approximately 900 SOMs were trained.

The training algorithm was implemented in Python, using the Numpy library (Harris et al., 2020) for array computations and the DTW implementation provided by the tslearn package (Tavenard et al., 2020). Training time per SOM ranged from 20 seconds for smaller maps (5×5 nodes) to 140 seconds for larger maps (13×13 nodes), with a total execution time of approximately 15 hours on an AMD Ryzen 9 PRO 5945 12-Core Processor (3.00 GHz). It is important to note that this computing time applies only to the training phase. Once trained, the SOM can classify hundreds of loops in a fraction of a second.

Optimal SOM selection, as illustrated in the fourth step of Fig. 5, was guided by our conceptual understanding of sediment transport loop types. For instance, maps were ranked higher when clockwise and counterclockwise *Figure L* loops were placed farther apart than clockwise and counterclockwise *Sync-Peak* loops (see Fig. 6), reflecting the greater temporal mismatch between discharge and turbidity in the former. This conceptual prioritization helped ensure that the SOM captured meaningful distinctions in loop behavior.

In the final refinement step, the selected SOM was retrained for an additional five epochs to enhance quantization accuracy. While the initial training emphasized the overall arrangement of loop types, this refinement focused on improving the alignment between each BMU and its

associated samples. To achieve this, a constant neighborhood spread of 0.45 and learning rate of 0.05 were employed.

**3.1.3 Evaluation**

We followed the principles defined in section 2.6 to evaluate the trained SOM's ability to represent the full spectrum of loop types in the curated dataset. Specifically, we examined whether loops from the same or similar categories (e.g., *single-lines*) were consistently mapped to prototypes that reflect their characteristic shapes. We also assessed the extent to which the SOM distinguishes between conceptually contrasting C–Q dynamics, such as clockwise versus counterclockwise *Figure L* loops, indicating its capacity to capture meaningful differences in loop behavior.

**3.2 Analysis phase**

With a trained and evaluated SOM for turbidity-discharge hysteresis, we now demonstrate the application of the SOM for analyzing hysteresis patterns using a secondary dataset consisting of T-Q hysteresis loops from three monitoring stations. The coordinates of these stations and key properties of their associated watersheds are provided in Table 1. Watershed boundaries were obtained from the USGS StreamStats online application, watershed slope was calculated using the USGS 3DEP (10m) digital elevation model, urban area was extracted from the 2021 National Land Cover Database, and soil texture was extracted from the Probabilistic Remapping of SSURGO (POLARIS) database (Chaney et al., 2019).

These watersheds were selected to illustrate two primary analyses for characterizing hysteresis patterns: (1) exploring the frequency distribution of loop types within a watershed (gages 07364130 and 03254480), and (2) identifying associations between loop types and hydrologic variables (gage 03289000).

Table 1. USGS monitoring stations and watershed properties used in the SOM-based analysis phase.

| USGS code                  |          | 07364130 | 03254480 | 03289000 |
|----------------------------|----------|----------|----------|----------|
| Latitude                   |          | 33.96    | 38.84    | 38.04    |
| Longitude                  |          | -91.69   | -84.53   | -84.63   |
| Watershed Area (km²)       |          | 311      | 47       | 62       |
| Watershed Mean Slope (m/m) |          | 0.012    | 0.13     | 0.08     |
| Watershed Urban Area (%)   |          | 2.03     | 1.31     | 15.8     |
| Mean Discharge (m³ s-1)    |          | 7        | 0.85     | 1.2      |
| Soil Texture               | Sand (%) | 9        | 6        | 6        |
|                            | Silt (%) | 50       | 61       | 64       |
|                            | Clay (%) | 41       | 33       | 30       |

Hydrologic events for watersheds 07364130, 03254480, and 03289000 were manually extracted from 15-minute discharge and turbidity data provided by the USGS and retrieved via the National Water Information System. For station 03289000, turbidity data were collected by the University of

Kentucky using a YSI 6-series optical turbidity sensor. All loops underwent preprocessing consistent with the curated dataset (see Section 3.1.1). This process resulted in 27, 54, and 70 events for gages 07364130, 03254480, and 03289000, respectively. Additional hydrologic variables were included for watershed 03289000, including rainfall 15 hours prior to events, average discharge over the 5 preceding days, and the ratio of old-water to event-water, derived from our previous work in this watershed (Marin-Ramirez et al., 2024). These variables serve as proxies for rainfall, antecedent moisture conditions, and dominant hydrologic pathways associated with each event, respectively.

For each site, loop frequency distributions were generated by mapping loops to their corresponding SOM prototypes and visualizing them as heatmaps. Distributions for 07364130 and 03254480 were compared to those obtained using the hysteresis index proposed by Zuecco et al. (2016). For watershed 03289000, the trained SOM was employed to explore relationships between loop types and hydrologic variables. This station was selected based on prior understanding of the hydrologic processes controlling sediment transport in the watershed (Marin-Ramirez et al., 2024), providing a framework for validation and comparison with the trained SOM. Associations between hydrologic variables and loop types were initially explored through visual analysis, where median values of the variables were plotted across SOM prototypes as heatmaps to reveal patterns linking high or low values with specific loop types.

To quantitatively assess these associations, a correlation approach was applied. First, BMU coordinates were transformed into a two-dimensional index (*row, column*). This two-dimensional index served as the predictor variable in a multiple linear regression model. Rank-normalized hydrologic variables (precipitation, antecedent discharge and the ratio of old-water to event-water) were used as predictands, allowing the model to capture non-linear, monotonic relationships. However, non-normalized variables could also be employed. Correlation coefficients were used to evaluate the strength of associations, while the regression plane's coefficients represented gradients showing the direction of maximum change in the hydrologic variable. These gradients helped identify loop types associated with higher or lower values of the variables. Results were visualized using a biplot chart, enabling simultaneous exploration of relationships between loop types and multiple hydrologic variables.